# Chemosensitization of prostate cancer stem cells in mice by angiogenin and plexin-B2 inhibitors

Shuping Li[1], Kevin A. Goncalves[1,2], Baiqing Lyu[1], Liang Yuan[1,3] & Guo-fu Hu [1,2,3]*

Cancer stem cells (CSCs) are an obstacle in cancer therapy and are a major cause of drug resistance, cancer recurrence, and metastasis. Available treatments, targeting proliferating cancer cells, are not effective in eliminating quiescent CSCs. Identification of CSC regulators will help design therapeutic strategies to sensitize drug-resistant CSCs for chemo-eradication. Here, we show that angiogenin and plexin-B2 regulate the stemness of prostate CSCs, and that inhibitors of angiogenin/plexin-B2 sensitize prostate CSCs to chemotherapy. Prostate CSCs capable of self-renewal, differentiation, and tumor initiation with a single cell inoculation were identified and shown to be regulated by angiogenin/plexin-B2 that promotes quiescence and self-renewal through 5S ribosomal RNA processing and generation of the bioactive 3'-end fragments of 5S ribosomal RNA, which suppress protein translation and restrict cell cycling. Monoclonal antibodies of angiogenin and plexin-B2 decrease the stemness of prostate CSCs and sensitize them to chemotherapeutic agents in vitro and in vivo.

[1] Division of Hematology-Oncology, Department of Medicine, Tufts Medical Center, Boston, MA, USA. [2] Graduate Program in Cellular and Molecular Physiology, Graduate School of Biomedical Sciences, Tufts University, Boston, MA, USA. [3] Program in Cellular, Molecular, and Developmental Biology, Graduate School of Biomedical Sciences, Tufts University, Boston, MA, USA. *email: guo-fu.hu@tufts.edu

Angiogenin (ANG) is a member of the pancreatic ribonuclease superfamily originally identified as a tumor angiogenic protein[1]. It is widely expressed[2] and has diverse biological functions ranging from cell growth and survival[3], neurogenesis and neuroprotection[4], innate immune reactions[5], and hematopoietic regeneration[6,7]. The growth-promoting activities of ANG, which are manifested by its robust tumor angiogenic and cancer progression effects[8], are mediated by promoting 47S ribosomal RNA (rRNA) transcription[9] through epigenetic activation of the ribosomal DNA promoter[10]. The survival function of ANG is mediated by producing a class of bioactive small RNA, the tRNA-derived stress-induced small RNA (tiRNA)[11,12], that suppresses global protein translation but permits internal ribosome entry sequence-mediated translation of pro-survival genes[13]. The differential RNA processing activities of ANG, which lead to distinct biological functions, have recently been shown in hematopoietic stem and progenitor cells (HSPCs) and myeloid progenitor cells (MyePros)[6]. We have reported that bone marrow niche-secreted ANG[7] promotes quiescence of primitive HSPCs by enhancing tiRNA production, which leads to restricted protein synthesis in these cells, whereas, in more differentiated MyePros, ANG promotes 47S rRNA transcription thereby stimulating protein synthesis and cell proliferation[6]. The differential functions of ANG in primitive stem cells and in differentiated cells are both mediated by plexin-B2 (PLXNB2), a recently identified functional ANG receptor that is both necessary and sufficient for the physiological and pathological functions of ANG in multiple cell types[14].

The dichotomous functions of ANG and PLXNB2 in HSPCs and MyePros prompted us to examine their function in prostate CSCs and in differentiated cancer cells, based on the rational that CSCs and HSPCs are regulated by similar mechanisms[15]. We hypothesized that while ANG stimulates the proliferation of differentiated prostate cancer cells, a function well-documented in the literature[8,16–22], it restricts the proliferation of prostate CSCs and preserves their self-renewal capacities by promoting quiescence. Since the original reports of prostate CSCs from three independent studies in 2005[23–25], accumulating evidence[26] points to the existence of prostate CSCs that are intrinsically resistant to antiandrogens, chemotherapeutic drugs, pro-oxidants, and radiation[27–29]. Numerous cell surface markers including aldehyde dehydrogenase (ALDH), CD133, CD44, CD24, and CD49f, have been reported to be associated with prostate CSCs[30–33]. However, until now, reliable cell surface markers that can be used to sort authentic prostate CSCs are still lacking[34]. A workshop of American Association of Cancer Research determined that the three criteria of authentic CSCs are self-renewal, differentiation, and tumor-initiating capacity[35], which were therefore used to identify putative human prostate CSCs by limited dilution method from established human prostate cancer cell lines. We obtained cell clones that are quiescent, able to self-renew, differentiate, and generate heterogeneous xenograft tumors in NOD scid gamma (NSG) mice with a single cell inoculation. The stemness of these cells is regulated by ANG and PLXNB2 through a mechanism involving ANG-dependent 5S rRNA processing and generation of bioactive 3′-end fragments of 5S rRNA. Inhibition of ANG and PLXNB2 by shRNA or monoclonal antibodies (mAbs) reduces the stemness of prostate CSCs, mobilizes them into cell cycle, and sensitizes them to chemotherapy. Combinatorial therapy of docetaxel (DTX) and a PLXNB2 mAb inhibited chemo-resistant CSC tumors and delayed disease recurrence, suggesting that targeting ANG/PLXNB2 is an effective means for eliminating quiescent CSCs in tumor therapy.

## Results

**Isolation and validation of prostate CSCs.** Single cells from PC3, DU145, and LNCaP human prostate cancer cell lines were cultured in 96-well non-adherent plates in sphere-forming medium for 4 months. The clones formed (Fig. 1a) were further screened by serial propagation for at least five passages (Fig. 1b). A total of 18 clones (six each from PC3, DU145, and LNCaP) were examined for their ability to initiate xenograft tumors in NSG mice. Among them, two clones from PC3 and one clone each from DU145 and LNCaP were able to form tumors in NSG mice with 100 cells and were thus further studied. Serial dilutions of these cells were inoculated and tumors were examined three months post inoculation. Tumor-take rate was 100% in mice inoculated with 100 cells of PC3 CSC1, DU145 CSC, and LNCaP CSC, and was 30% with 100 cells of PC3 CSC2 (Table 1). Two of the six mice inoculated with a single CSC1 of PC3 developed tumors, and six of the eight mice developed tumors when inoculated with a single CSC of DU145 and LNCaP (Table 1). No tumors were detected in mice inoculated with one or ten cells of CSC2 line of PC3 (Table 1), suggesting considerable clonal heterogeneity of CSCs. CSC1 of PC3 was used throughout the study and CSC2 was no longer studied. Under the same conditions, at least 50,000 parent PC3 cells were required to initiate xenograft tumors in NSG mice.

The ability of these cells to form spheres was greatly enhanced as compared with their respective parent cells (Fig. 1c). The prostatospheres were identified morphologically as structures with clear membrane-like circle boundaries and were differentiated from cell aggregates that displayed a polymorphic structure. The number of spheres formed from CSCs of PC3, DU145, and LNCaP was 44.6-, 53.6-, and 48.6-fold over that from the same numbers of the respective parent cells, respectively (Fig. 1c). Similar results were obtained in limited dilution analysis (Fig. 1d). No appreciable decrease in sphere-forming ability was noted for at least five passages in serial replating experiments (Fig. 1e). These data suggest that the CSCs have enhanced self-renewal ability as it has been demonstrated that only-self-renewing cells are capable of maintaining their sphere-forming potential in multiple generation[27].

Flow cytometry analysis showed that the G0 cell frequency of CSCs cloned from PC3, DU145, and LNCaP cells were 5.5-, 3.4-, and 8.7-fold of that of the respective parent cells, respectively (Fig. 1f), indicating that the CSCs were quiescent. Protein synthesis is tightly regulated in stem cells[36] and has been shown to be closely associated with HSPC stemness[6]. We examined protein synthesis rates of the three CSC lines using O-propargyl-puromycin (OP-Puro) incorporation[6,36] and found that protein synthesis rate was universally lower in CSCs than in their respective parent cells (Fig. 1g), confirming the stemness property of these CSCs.

Consistent with the quiescent status and a low protein synthesis rate, CSCs have reduced proliferation rates as compared with their respective parent cells. They proliferated slower in vitro than the parent cells until day 40 in culture (Fig. 1h) with the biggest difference observed in the early phase of culture. The difference in proliferation rate between CSCs and parent cells of PC3 gradually decreased in a prolonged culture and reversed by day 40, when the parent cells reached a plateau but CSCs remained proliferating, a phenomenon that has been previously observed[37]. Tumors initiated from CSCs also grew slower in vivo than did those initiated from an equal number of parent cells (Fig. 1i) before they picked up speed around week 2 (Fig. 1j). Similar growth characteristics were also observed in CSCs of DU145 and LNCaP cells (Supplementary Fig. 1a, b). These data demonstrate that CSCs are metabolically active and are not senescent, and are able to proliferate and differentiate in vitro and in vivo.

We also found that CSCs have enhanced bone marrow tropism and ability to compete with HSPCs for bone marrow niche residency as compared with parent cells. We transplanted human CD34+ cord blood cells into sub-lethally irradiated NSG mice, and confirmed successful engraftment of both human and mouse cells in the bone marrow 16 weeks post transplantation. BM cells from

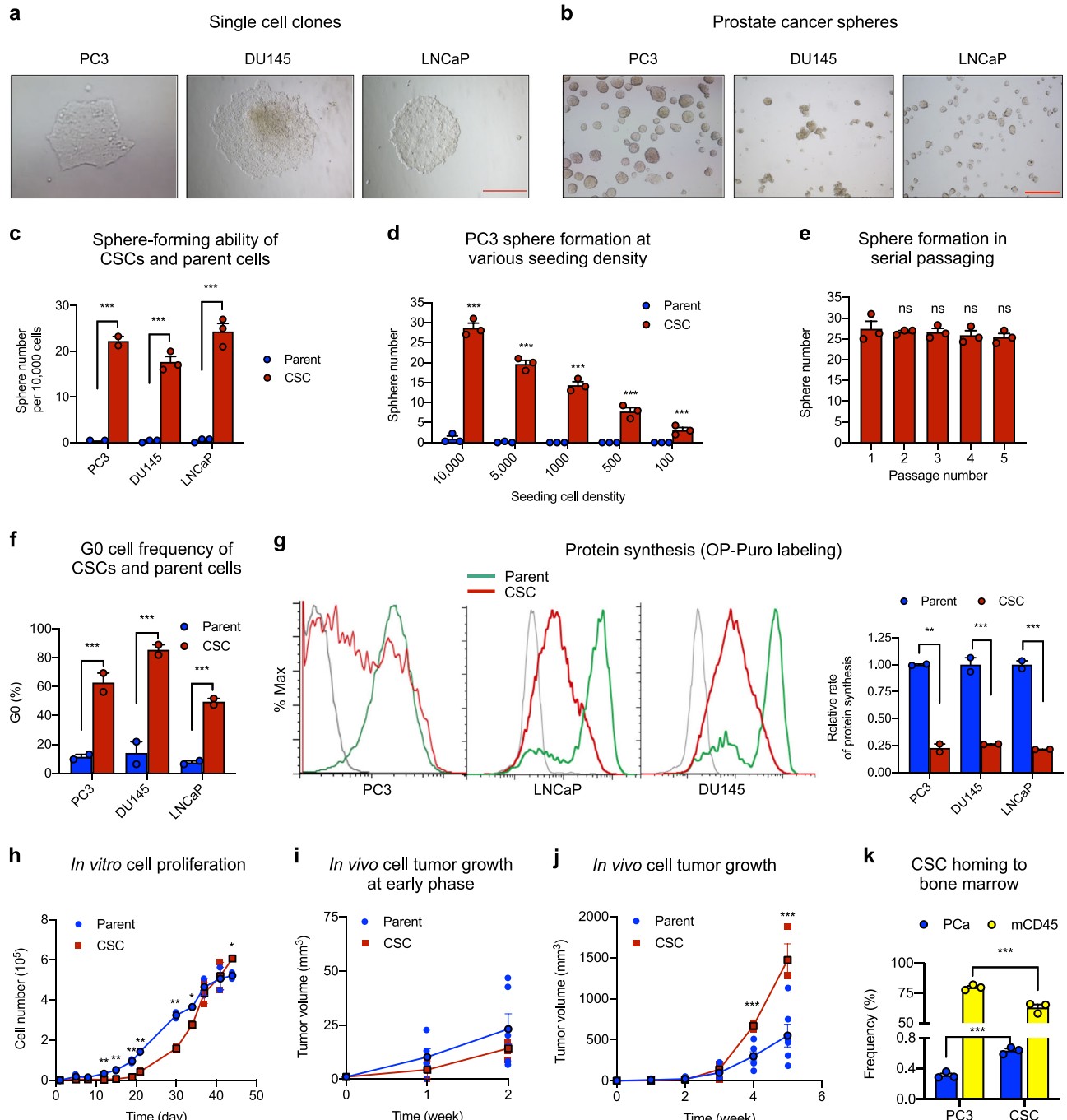

**Fig. 1 Isolation and characterization of prostate CSCs. a** Morphology of single cell clones isolated from PC3, DU145, and LNCaP. Scale bar: 500 μm. **b** Cancer sphere formed from CSCs cultured in sphere medium in bacteriological petri dishes. Scale bar: 500 μm. **c** Sphere formation of the CSC lines and their respective parent cells at seeding density of 10,000 per 2 ml in 35-mm dishes ($n = 3$). **d** Sphere formation of PC3 CSCs and parent cells at 100–10,000 cells per 2 ml in 35-mm dishes ($n = 3$). **e** Sphere formation during serial passaging (10,000 cells per 2 ml in 35-mm dishes) ($n = 3$). **f** G0 cell population of CSCs and parent cell lines determined by flow cytometry after Ki-67 and 7-AAD staining ($n = 3$). **g** Protein synthesis rate in CSCs and parent cells determined by OP-Puro incorporation followed by flow cytometry analysis ($n = 3$). Geometric mean was calculated by FlowJo. Bar graphs representing relative protein synthesis rate between CSCs and respective parent cells. **h** In vitro growth curves of CSCs and parent cells of PC3. Seeding density was 10,000 cells per well of 48-well plates ($n = 3$). Early (**i**) and late (**j**) phase of in vivo growth of PC3 CSCs and parent cells in NSG mice. A mixture of 50 μl HBSS containing 50,000 cells and 50 μl Matrigel was inoculated in NSG mice ($n = 8$). Tumor size was measured every week. **k** Bone marrow niche binding of PC3 cells and CSCs. GFP-labeled PC3 cells or CSCs were administered to NSG mice that had been transplanted with total bone marrow mononuclear cells (BMMNC) from the first recipient mice that had been transplanted with human CD34[+] cells for 16 weeks. Frequency of GFP positive cells and total mouse CD45 cells in the BM was determined by flow cytometry 2 weeks post administration of cancer cells ($n = 6$). *$p < 0.05$; **$p < 0.01$; ***$p < 0.001$; ns not significant.

**Table 1 Tumor-initiating capacity of candidate prostate CSCs.**

| Number of cells | Tumor-bearing mice/total mice | | | |
| --- | --- | --- | --- | --- |
| | PC3 CSC1 | PC3 CSC2 | DU145 CSC | LNCaP CSC |
| 1 | 2/8 | 0/8 | 6/8 | 6/8 |
| 10 | 3/10 | 0/8 | NA | NA |
| 100 | 10/10 | 3/10 | 8/8 | 8/8 |
| 1000 | 10/10 | 10/10 | NA | NA |
| 5000 | 8/8 | 8/8 | NA | NA |

CSCs of PC3, DU145, LNCaP ranging from 1 to 5,000 cells were inoculated in NSG mice ($n$ = 8–10). Tumors were examined 3 months post inoculation
NA not assayed

the above primary recipient mice were used as donor cells for the secondary transplantation to ensure a more homogenous engraftment among the recipients. Two weeks after the secondary transplantation, GFP-labeled PC3 parent cells or CSCs were intravenously administered and BM was analyzed after another 4 weeks for mouse CD45 cells and GFP positive cancer cells. More CSCs have engrafted to the BM, as compared with parent cells, resulting in a decrease of mouse cell engraftment (Fig. 1k), indicating that CSCs have enhanced BM niche binding capacity as compared with differentiated cancer cells. No GFP-labeled parent cells or CSCs were detected in other organs including lungs and lymph nodes in these animals under this condition.

Consistent with the undifferentiated nature of stem cells, the three prostate CSC lines have a decreased expression of basal cell markers CK5 (*KRT5*) and CK14 (*KRT14*), and luminal marker CK18 (*KRT18*) as shown by qRT-PCR (Fig. 2a) and immunofluorescence (Supplementary Fig. 2). The observation of a universal decrease in CK5, CK14, and CK18 expression in the three CSC lines as compared with their corresponding parent cells is in contrast to a previous report that CSCs with the phenotype of ALDH$^+$ CD44$^+$α2β1$^+$ had high expression of CK5 and CK14[38], suggesting the heterogeneous nature of the CSCs. No consistent expression patterns of the intermediate cell markers were noted as CK19 (*KRT19*) was decreased in CSCs of PC3 but enhanced in CSCs of DU145 and LNCaP, whereas glutathione-S-transferase-pi (*GSTP1*) was decreased in CSCs of DU145 but increased in CSCs of PC3 and LNCaP. However, at least one of the two neuroendocrine markers was elevated in all three CSC lines: synaptophysin (*SYP*) was enhanced in CSCs of PC3, chromogranin-A (*CHGA*) was elevated in CSCs of DU145, and both synaptophysin and chromogranin-A were elevated in CSCs of LNCaP (Fig. 2a and Supplementary Fig. 2).

Next, we examined the differentiation potential of the candidate CSCs as it has been shown that the plasticity of CSCs endows them to differentiate into heterogeneous lineages of cancer cells[37]. While the candidate CSCs maintained their sphere-forming capability and remained undifferentiated when cultured in sphere medium in non-adherent dishes, they underwent differentiation and subsequent proliferation in the presence of serum under adherent culture conditions as evidenced by the formation of large colonies and outward growth of cells (Fig. 2b), accompanied by enhanced expression of basal, luminal, and neuroendocrine markers (Fig. 2c, Supplementary Fig. 3a). We also found that these candidate CSCs were able to differentiate in vivo. Tumors derived from both CSCs and parent cells of PC3 were heterogeneous, and had enhanced expression of basal, luminal, and neuroendocrine markers than their respective originating cells (Fig. 2d). As a result, tumors derived from CSCs and parent cells of PC3 displayed a similar level of CK18 (Fig. 2e) and SYP (Fig. 2f), indicating that both parent cells and CSCs were able to form differentiated tumors in mice.

Androgen receptor (AR) has been reported to regulate prostate cancer stemness[39]. AR mRNA and protein are detectable in both androgen-sensitive LNCaP cells and -insensitive DU145 and PC3 cells[40]. In the candidate CSCs cloned from androgen-sensitive LNCaP cells, we found that *AR* mRNA level was dramatically lower than in parent cells, however, no consensus expression pattern of *AR* was found in the candidate CSCs cloned from androgen-insensitive cell lines PC3 and DU145 (Supplementary Fig. 3b), consistent with the finding that AR could be either positive or negative in prostate CSCs, and confirming the heterogeneous level and diverse function of AR in stem cells[41,42]. We also found that AR was reexpressed in xenograft tumors derived from both CSCs (Supplementary Fig. 3c) and parent cells (Supplementary Fig. 3d) of PC3, further demonstrating that the candidate CSCs are able to differentiate in vivo to generate tumors with features of prostate cancer. A decrease in prostate specific antigen (PSA) level has been reported as an indicator of stemness of prostate cancer cells[43]. However, no difference in PSA levels was found between CSCs (0.62 ± 0.09 ng per 10$^3$ cells per day) and parent cells (0.65 ± 0.07 ng per 10$^3$ cells per day) of LNCaP.

Taken together, these results demonstrate that the candidate CSCs cloned form prostate cancer cells lines are quiescent, able to self-renew, differentiate, and initiate tumors with a single cell inoculation, indicating that they are authentic prostate CSCs. To the best of our knowledge, this is the first time that a single cell is shown to be able to generate serially transplantable and fully differentiated tumors in mice, even though one previous report showed that single luminal epithelial progenitors can generate prostate organoids in culture[44].

**ANG and PLXNB2 regulate stemness of prostate CSCs.** Cell surface markers including ALDH, CD24, CD44, CD49f, CD133, CD326, and Trop2, have been used to sort potential prostate CSCs[30–33]. We analyzed the expression patterns of these cell surface molecules in the CSCs cloned from PC3, DU145, and LNCaP cell lines by flow cytometry and found that all three CSC lines have increased expression of CD49f (Supplementary Fig. 4a) and decreased expression of CD133 (Supplementary Fig. 4b). No consensus expression patterns of ALDH, CD24, CD44, CD326, and Trop2 were noticed among the three CSC lines (Supplementary Fig. 4c–4g).

We next performed a qPCR analysis to examine the expression level of human stem cell transcription factors in CSCs and parent cells of PC3. Among the 88 transcription factors analyzed, *GATA3*, *SOX2*, and *MYC* were the three transcription factors that displayed elevated expression in CSCs as compared with parent cells. These three transcription factors, as well as a number of self-renewal and cycle checkpoint genes including, *ITGA2*, *MUC1*, *BMP7*, *DNMT1*, *KLF4*, *CLDN4*, *CDKN1B*, and *CDKN1A*, which have been reported to be associated with stemness, were analyzed in all three CSC lines and found that they were overall higher in CSCs as compared with their respective parent cells (Fig. 3a). It is notable that *CCND1*, the cell cycle-related gene, was lower in CSCs than in parent cells, in agreement with reduced cell cycle status of CSCs.

ANG has been shown to be progressively upregulated in prostate cancer[8,16,17,19,45] and plays a role in the transition from androgen-dependent to castration-resistant, hormone-refractory phenotype[21,45]. *PLXNB2* has also been shown to be upregulated in prostate cancer and be inversely correlated with patient survival[14]. Expression levels of ANG (Supplementary Fig 5a) and PLXNB2 (Supplementary Fig. 5b,5c) are higher in CSCs than in parent cells. Knockdown of *ANG* by ANG-specific shRNA E7 and E4 (Fig. 3b, Supplementary Fig. 6) or *PLXNB2* by

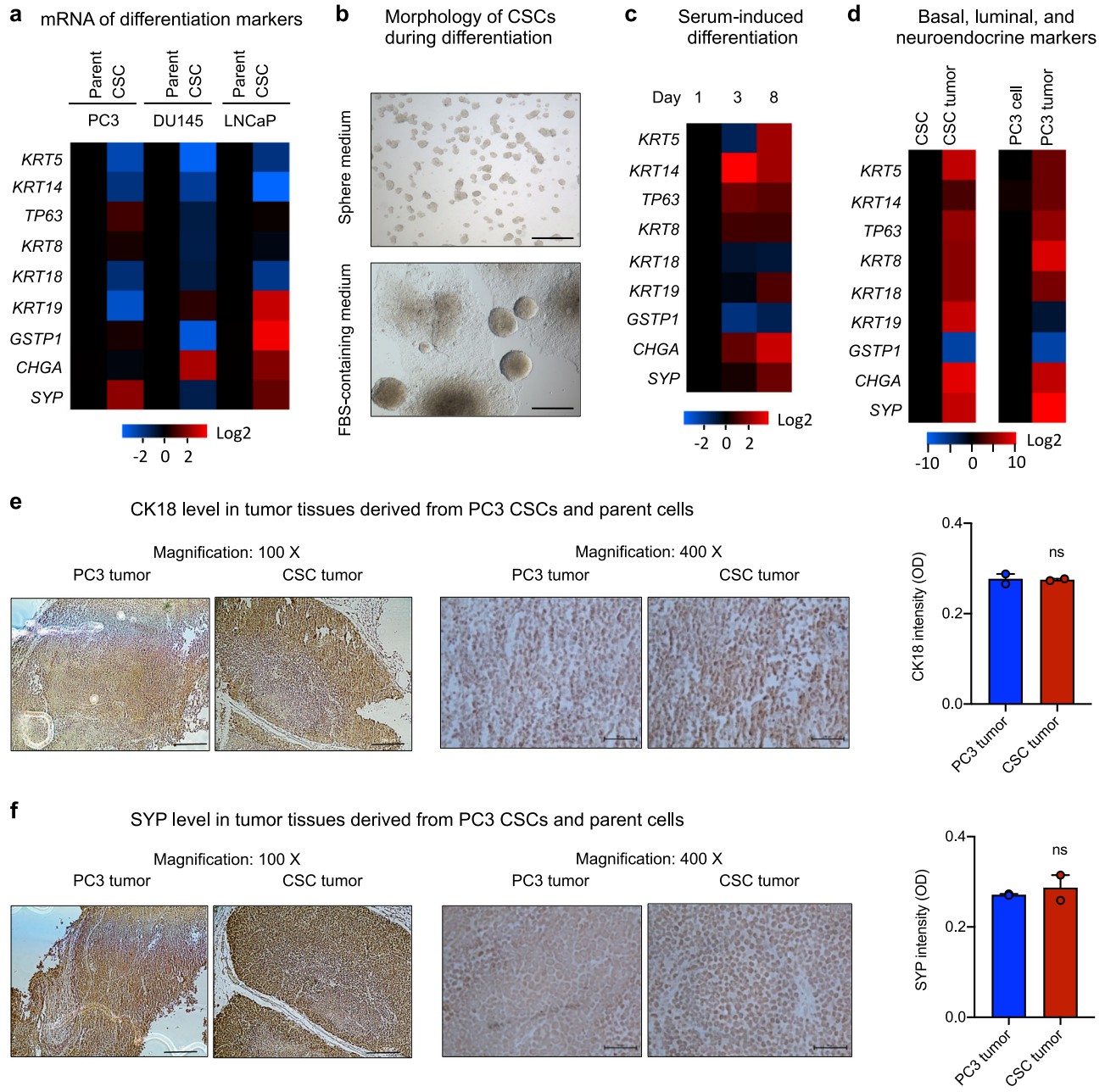

**Fig. 2 Differentiation potential of prostate CSCs. a** Expression levels of basal, luminal, and neuroendocrine markers in CSCs and parent cells measured by qRT-PCR ($n = 3$). Values in the CSCs were normalized to the respective parent cells. **b** Morphology of PC3 CSCs cultured in sphere medium in non-adherent petri dish (top) and in DMEM plus 10% FBS in cell culture dish (bottom) for 2 weeks. Scale bar, 500 μm. **c** mRNA levels of basal, luminal, and neuroendocrine markers in CSCs cultured in regular cell culture medium (DMEM + 10% FBS) for 3 and 8 days ($n = 3$). Heatmaps represent the relative mRNA level of each marker normalized to the value at day 1. **d** mRNA levels of basal, luminal, and neuroendocrine markers in CSCs and CSC-derived tumors (left), and in PC3 cells and PC3 cell-derived tumors (right) ($n = 3$). Values in the tumor tissues were normalized to the originating cells. IHC of CK18 (**e**) and SYP (**f**) in tumors derived from PC3 cells or CSCs. Quantitation of signal intensity was obtained by Image J ($n = 3$). Scale bar: left panels, 200 μm; right panels, 50 μm. ns not significant.

PLXNB2-specific shRNA 489 and 549 (Fig. 3c, Supplementary Fig. 6) in CSCs resulted in a reduction of stemness gene expression (Fig. 3d), a decrease in G0 cell frequency with a concomitant increase in the frequency of S-G2-M cells (Fig. 3e, f), and a decrease in sphere-forming capability (Fig. 3g, h). Consistently, *ANG* and *PLXNB2* knockdown CSCs proliferated faster both in vitro (Supplementary Fig. 7a,b) and in vivo (Supplementary Fig. 7c). Tumors derived from *ANG* and *PLXNB2* knockdown CSCs were larger than those derived from the same number of control shRNA-transfected CSCs, accompanied with an increase

in Ki-67 positive cells in the tumor sections (Supplementary Fig. 7d). No difference was noted in spheroid invasion[46] of *ANG* and *PLXNB2* knockdown CSCs (Supplementary Fig. 7e). Importantly, we found that *ANG* and *PLXNB2* knockdown resulted in a faster exhaustion of the in vivo tumor-initiating ability of CSCs as assessed by the relative tumor growth rate between passages one and three (Fig. 3i). These results demonstrate that a decrease in *ANG* or *PLXNB2* expression in CSCs led to enhanced cell cycling and reduced self-renewal capacity, suggesting that ANG and PLXNB2 maintain CSC stemness, likely by keeping them in

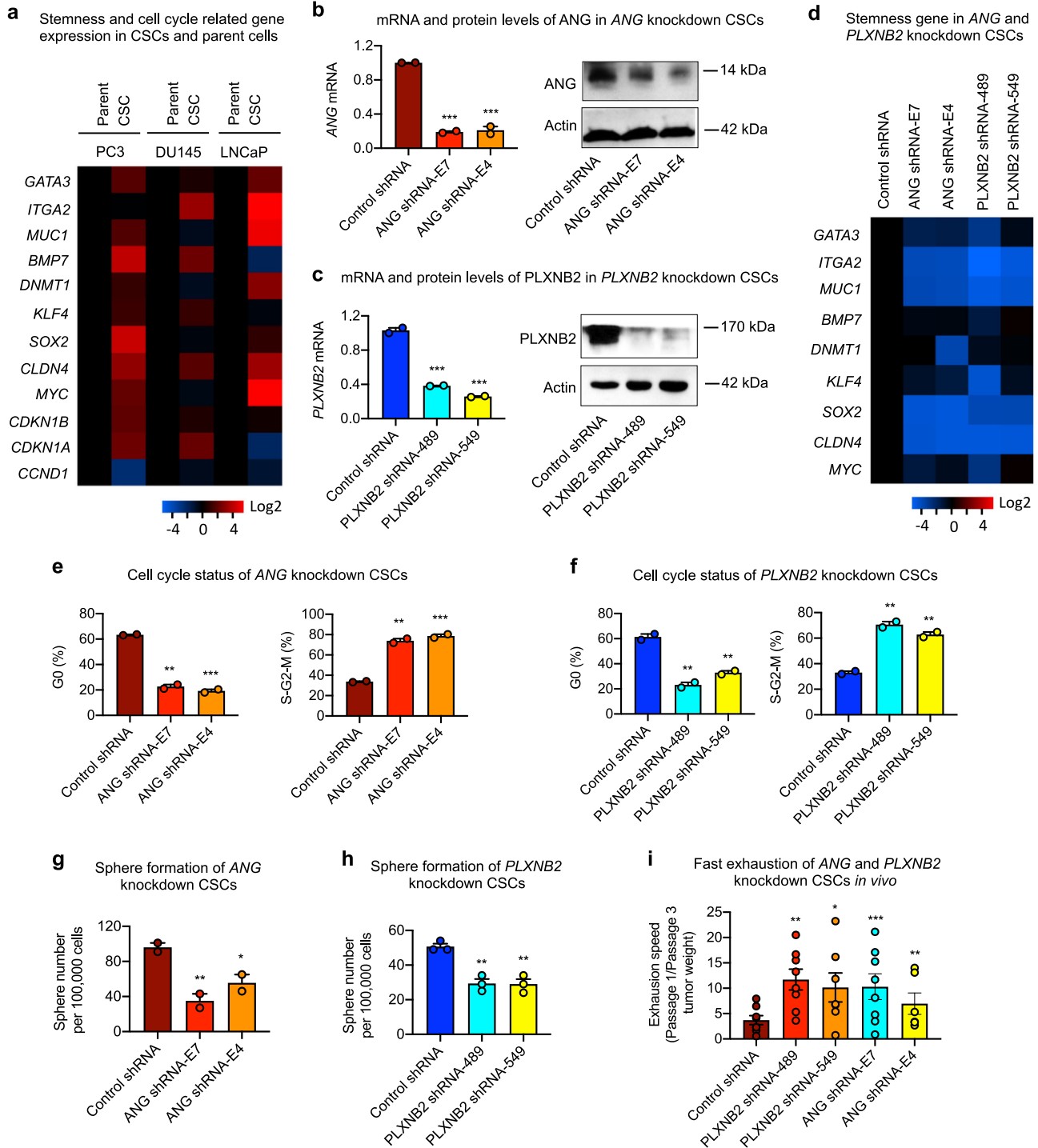

**Fig. 3 Knockdown of *ANG* or *PLXNB2* decreases stemness of prostate CSCs. a** mRNA levels of cancer stemness-related genes in prostate CSCs and parent cells (*n* = 5). Values in the CSCs were normalized to the respective parent cells. mRNA and protein levels of ANG (**b**) and PLXNB2 (**c**) in knockdown CSCs. mRNA levels were determined by qRT-PCR and normalized to control shRNA transfectants (*n* = 3). Protein levels were determined by immunoblotting. **d** mRNA level of cancer stemness-related genes in *ANG* and *PLXNB2* knockdown CSCs (*n* = 3). Values were normalized to the control shRNA transfectants. Cell cycle status of *ANG* (**e**) and *PLXNB2* (**f**) knockdown CSCs, analyzed by flow cytometry after Ki-67 and 7-AAD staining (*n* = 2). Sphere formation of *ANG* (**g**) and *PLXNB2* (**h**) knockdown CSCs (*n* = 3). **i** CSC exhaustion during serial passaging in vivo. Cells were passaged in NSG mice (*n* = 5–8) for three times. In each passage, 100,000 cells were inoculated per mouse. Tumors were excised and weighed 4 weeks post inoculation in each passage. Exhaustion rate was calculated as the ratio of tumor weight from first passage to third passage. *$p < 0.05$; **$p < 0.01$; ***$p < 0.001$.

quiescence, as unchecked stem cell proliferation has been demonstrated to result in exhaustion[47].

As PLXNB2, CD49f, and ALDH are overexpressed in CSCs than in parent cells of PC3, we sorted PLXNB2$^{high}$CD49f$^{high}$ALDH$^{high}$ cells from PC3 cells and examined their ability to form prostato-spheres. However, PLXNB2$^{high}$CD49f$^{high}$ALDH$^{high}$ cells displayed no enhancement in sphere-forming capabilities as compared with the parent cells (Supplementary Fig. 7f), indicating that *PLXNB2* is

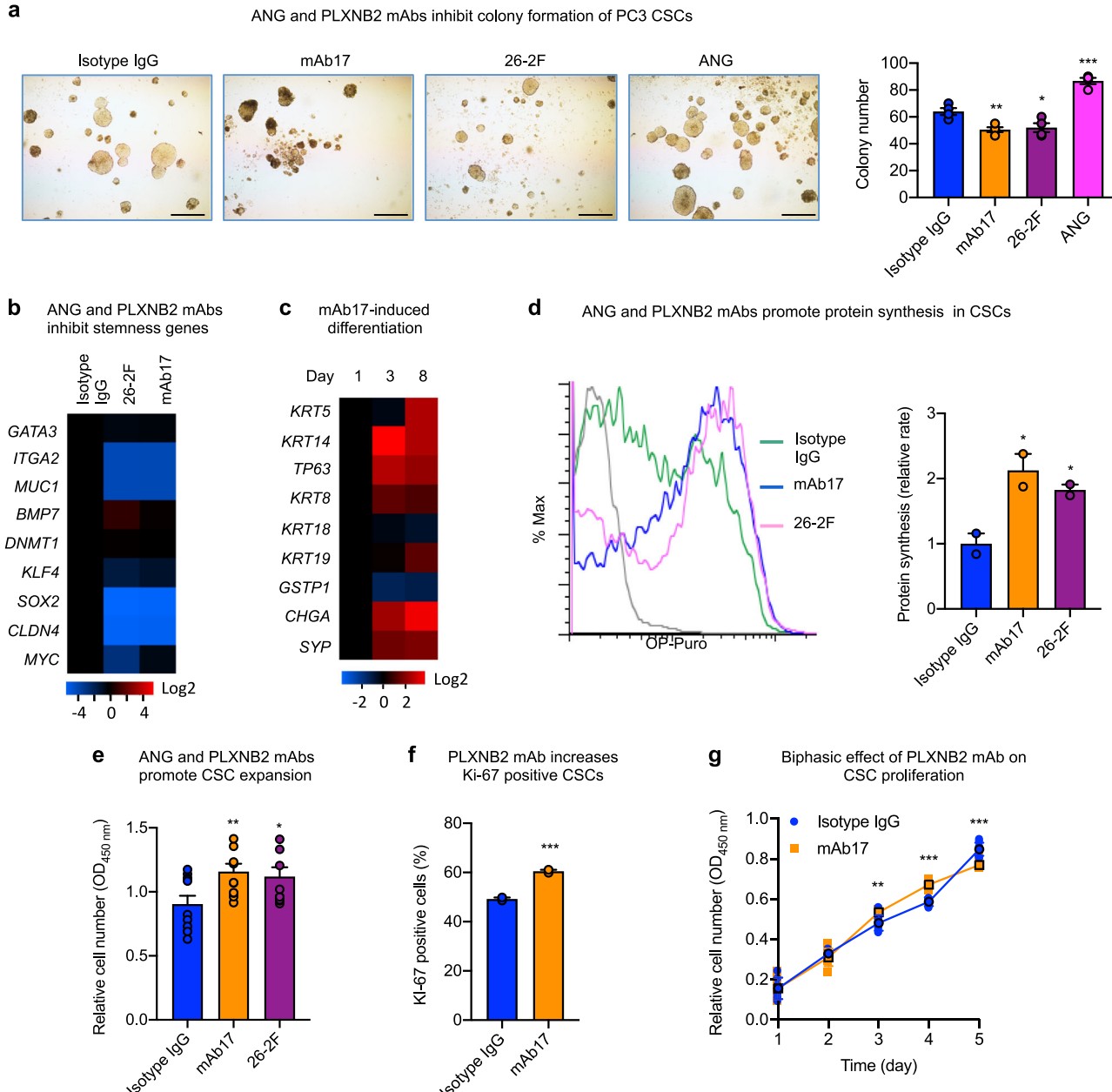

**Fig. 4 ANG and PLXNB2 mAbs decrease stemness of prostate CSCs. a** Colony formation of PC3 CSCs in the presence of 30 μg/ml of non-immune isotype IgG control, 26–2F, or mAb17, or 1 μg/ml of ANG ($n = 3$). Scale bar: 500 μm. **b** mRNA levels of stemness genes in CSCs treated with 30 μg/ml isotype IgG, mAb17, or 26–2F for 24 h ($n = 3$). Values were normalized to isotype IgG control. **c** mRNA levels of basal, luminal, and neuroendocrine markers in CSCs on day 1, 3, and 8 in the presence of 30 μg/ml mAb17 ($n = 3$). Values were normalized to day 1. **d** Protein synthesis in CSCs treated with 30 μg/ml isotype IgG, mAb17, or 26–2F for 24 h determined by OP-Puro incorporation followed by flow cytometry. Geometric means were calculated and normalized to that of isotype IgG control ($n = 2$). **e** CSC proliferation in the presence of 30 μg/ml isotype IgG, mAb17, or 26–2F. Cell numbers were determined by MTS assay after 4 days of treatment ($n = 10$-11). **f** Ki-67 positive cell population determined by flow cytometry in CSCs after treatment with 30 μg/ml of isotype IgG control or mAb17 for 3 days ($n = 2$). **g** In vitro growth curve of PC3 CSCs in the presence of 30 μg/ml isotype IgG control or mAb17 ($n = 4$-8). $*p < 0.05$; $**p < 0.01$; $***p < 0.001$.

upregulated in CSCs but is insufficient to be used for sorting potential CSCs.

**ANG and PLXNB2 mAbs decrease stemness of prostate CSCs.** To determine whether ANG-regulated CSC quiescence is of therapeutic relevance, we examined the effect of mAbs of ANG and PLXNB2 on CSC stemness. ANG mAb (26-2F) and PLXNB2 mAb (mAb17) inhibited colony formation of all three prostate CSC lines (Fig. 4a, Supplementary Fig. 8a,b), demonstrating that these antibodies decreased the functional capabilities of CSCs.

Expression of stemness-related genes was decreased upon treatment with 26-2F or mAb17 (Fig. 4b), whereas the expression of basal, luminal, and neuroendocrine markers was generally enhanced in a time-dependent manner by mAb17 (Fig. 4c), indicating that the neutralization of ANG or PLXNB2 by mAbs was able to decrease the stemness of CSCs and induce differentiation. Loss of CSC quiescence was accompanied by elevated protein synthesis in CSCs (Fig. 4d), resulting in CSC expansion (Fig. 4e). Consistently, mAb17 increased the frequency of Ki-67 positive cells (Fig. 4f), and had a biphasic effect on CSC

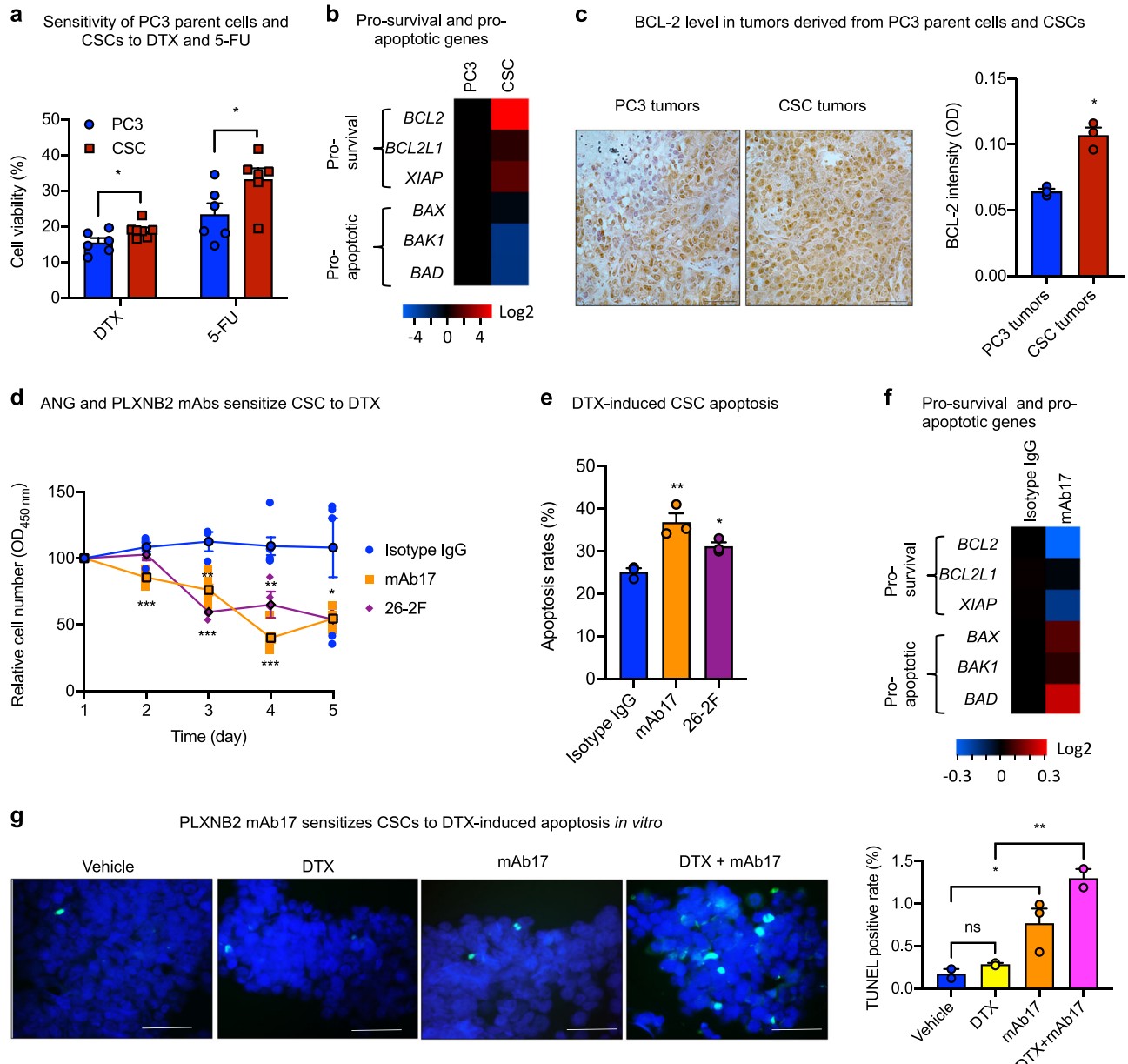

**Fig. 5 ANG and PLXNB2 mAbs sensitize prostate CSCs to chemotherapy. a** Viability of PC3 cells and CSCs treated with 50 nM DTX or 100 μM 5-FU for 24 h (n = 6). **b** Expression of pro-survival and pro-apoptotic genes in PC3 cells and CSCs (n = 3). Values were normalized to PC3 cells. **c** IHC of BCL-2 in xenograft tumor tissues derived from PC3 cells and CSCs. Signal intensity was determined by Image J (n = 3). **d** Time course of CSC viability in the presence of 20 nM DTX plus 30 μg/ml of isotype IgG control, mAb17, or 16-2 F (n = 4–6). **e** Apoptotic cells determined by flow cytometry followed by Annexin V staining in CSCs treated with 50 nM DTX in the presence of 30 μg/ml isotype IgG control, mAb17, or 26-2 F for 24 h (n = 3). **f** mRNA levels of pro-survival and pro-apoptotic genes in CSCs after treatment with 30 μg/ml of isotype IgG control or mAb17 for 48 h (n = 3). **g** Apoptotic cells detected by TUNEL assay in CSCs treated with 20 nM of DTX, 30 μg/ml mAb17, or both for 24 h. Scale bar: 50 μm. TUNEL positive cells were counted in a total of 200 cells in each sample (n = 3). *p < 0.05; **p < 0.01; ***p < 0.001; ns not significant.

proliferation (Fig. 4g). This biphasic effect of mAb17 in CSC culture is likely caused by the dichotomous effect of ANG on CSCs and on differentiated cancer cells. In the early phase of CSC culture, mAb17 likely mobilized CSCs out of quiescence and therefore enhanced their proliferation; however, in the later phase of culture, the population became more predominantly differentiated cancer cells due to CSC differentiation, mAb17 then likely inhibited proliferation of these differentiated cancer cells. This observation is consistent with the finding that treatment of PC3 parent cells with 26-2F and mAb17 resulted in reduced protein synthesis (Supplementary Fig. 8c) and slowed proliferation (Supplementary Fig. 8d, e). These results indicate that ANG

and PLXNB2 mAbs promote CSC cycling, enhance CSC proliferation and differentiation, and inhibit differentiated cancer cells.

**ANG and PLXNB2 mAbs sensitize prostate CSCs to chemotherapy.** CSCs were more resistant to chemotherapy drugs such as DTX and fluorouracil (5-FU) (Fig. 5a), likely as a result of enhanced expression of pro-survival genes and decreased expression of pro-apoptotic genes (Fig. 5b). Indeed, the enhanced expression of BCL-2 protein level was observed in tumors derived from CSCs than those derived from PC3 parent cells (Fig. 5c). DTX, the most commonly used chemotherapeutic agent for late

**Table 2 PLXNB2 mAb decreases IC50 of DTX in prostate CSCs3.**

| Cells | IC50 of DTX (nM) | |
|---|---|---|
| | Isotype IgG | mAb17 |
| PC3 | 12.7 | 12.7 |
| CSCs | 240 | 62 |

IC50 of DTX in CSCs and parent cells of PC3 in the presence of 30 µg/ml isotype IgG control or mAb17 for 24 h. IC50 was calculated as the concentration of DTX at which 50% of cell growth inhibition was achieved ($n = 6$)

stage prostate cancers, at 20 nM, a concentration comparable to the dose used in chemotherapy, did not affect CSC viability after 5 days but resulted in ~50% death when 30 µg/ml of 26-2F or mAb17 was present (Fig. 5d). At a high dose of 50 nM, DTX induced only $25.2 \pm 1.5\%$ of CSCs into apoptosis in 24 h, however, in the presence of 26-2F and mAb17, the same concentration of DTX induced $36.8 \pm 3.7\%$ and $31.2 \pm 1.6\%$ of CSCs into apoptosis, respectively (Fig. 5e), indicating that inhibition of ANG or PLXNB2 sensitized CSCs to DTX-induced apoptosis. Accordingly, mAb17 was found to attenuate pro-survival gene expression but enhance pro-apoptotic gene expression (Fig. 5f), leading to an increase in apoptosis (Fig. 5g). The synergistic effect of DTX and mAb17 on CSC apoptosis led to a decrease of IC50 of DTX toward CSCs from 240 to 62 nM (Table 2). It is notable that the IC50 of DTX toward PC3 parent cells was not affected by mAb17.

Next, we examined the chemo-sensitizing effect of PLXNB2 mAb in vivo. NSG mice bearing CSC xenograft tumors were treated with mAb17, DTX, or a combination of the two. mAb17 and DTX at 10 mg per kg each marginally inhibited CSC tumor growth (Fig. 6a), and displayed an additive effect when used together (Supplementary Fig. 9a). At a high dose (30 mg per kg), DTX had a greater inhibition and was further enhanced by mAb17, resulting in over 90% inhibition in tumor growth in the combinatorial treatment group (Fig. 6a). The degree of inhibition in the combinatorial treatment group was higher than the theoretically additive value of the two individual treatment groups (Supplementary Fig. 9b), indicating a synergistic effect between mAb17 and DTX. To determine if a combination treatment of DTX and mAb17 delayed tumor recurrence, individual mice bearing similar sizes of tumors from the DTX (30 mg per kg) group and from the DTX (30 mg per kg) + mAb17 (4.8 mg per ml) group were selected for post treatment observation of tumor development. Tumors in mice treated with DTX alone recurred much faster than that in mice treated with DTX and mAb17 (Fig. 6b), indicating that mAb17 not only sensitized CSC tumors to chemotherapy, but also delayed tumor recurrence after treatment was ceased. Recurred tumors from DTX + mAb17 group had slightly enhanced *ANG* expression as compared with those from DTX group, probably as a feedback response from PLXNB2 inhibition by mAb17 (Fig. 6c). However, no difference was observed in recurred tumors between DTX and DTX + mAb17 groups in the expression levels of *PLXNB2* and CSC markers including *ITGA6* (CD49f), *PROM1* (CD133), *CD24*, and *CD44* (Fig. 6c), suggesting that delayed tumor recurrence in the combinatorial treatment group was not due to changes of clonal composition induced by PLXNB2 inhibition. Consistent with in vitro findings that mAb17 promoted CSC differentiation and enhanced CSC apoptosis, especially in the presence of DTX, we found that CSC tumors from mAb17-treated mice have enhanced expression of basal, luminal, and neuroendocrine markers such as CK5, CK18, and SYP (Fig. 6d, Supplementary Fig. 9c) and apoptotic mediators such as cleaved PARP and caspase 6 (Fig. 6e, Supplementary Fig. 9d), and underwent more robust apoptosis

(Fig. 6f). TUNEL staining showed that tumor tissues from animals receiving combinatorial treatment of DTX and mAb17 had increased apoptosis over those receiving treatment with vehicle, DTX or mAb17 alone (Fig. 6f). Consistent with previous reports that inhibition of PLXNB2 by mAb17 had no adverse effect on normal tissues[14], no acute or delayed toxicity was noticed in mAb17-treated NSG mice.

Together, the above results demonstrate that ANG and PLXNB2 mAbs were able to decrease the self-renewal capacity of prostate CSCs and sensitize them to chemotherapy. It should be noted that all three CSC lines cloned from three independent parent cell lines were responsive to ANG or PLXNB2 mAbs, indicating that these CSCs are not random or peculiar clones that happen to be more tumorigenic, but rather share a common regulatory mechanism in which the ANG-PLXNB2 axis maintains their stemness.

**ANG promotes 5S rRNA processing in CSCs.** In order to understand the mechanism by which ANG maintains CSC quiescence, we first examined differential subcellular distribution of ANG in CSCs and parent cells of PC3, and found that while ANG was translocated to nucleus and accumulated in the nucleolus in PC3 parent cells, as has been reported previously[8], it was mainly located in stress granules and co-localized with poly (A)-binding protein (PABP), a stress granule marker, in CSCs (Supplementary Fig. 10). Stress granule accumulation of ANG has been shown to be associated with tiRNA production and is the underlying mechanism by which ANG restricts protein synthesis and cell proliferation of HSPCs[6]. However, while we indeed found that CSCs produce a larger amount of small RNAs (<50 nt) as compared with their parent cells (Fig. 7a, Supplementary Fig. 11), which could be further enhanced by exogenous ANG (Fig. 7b, Supplementary Fig. 11), we did not observe a detectable amount of tiRNA Gly$^{GCC}$, a type of tiRNA that has been shown to be responsible for ANG-enhanced self-renewal and quiescence of HSPCs[6], in prostate CSCs (Fig. 7c, Supplementary Fig. 11).

In order to determine which small RNA is responsible for ANG-regulated prostate CSC quiescence, we sequenced CSC-produced small RNAs and found that, among the 63 readable clones, 21 contained sequences from the 5S rRNA (Table 3, Supplementary Table 1). It is notable that most of the cleavages occurred at sites located in the β-domain of 5S rRNA (Supplementary Fig. 12), suggesting that the 3′-end fragment of the 5S rRNA remained relatively intact and might thus be functional. We therefore examined the abundance of small RNAs containing the 3′-end sequence of 5S rRNA by northern blotting and found that the amount of 3′-end fragments of 5S rRNA was higher in all three CSC lines as compared with the respective parent cells (Fig. 7d, Supplementary Fig. 11). Knockdown of either *ANG* or *PLXNB2* decreased the level of the 3′-end fragment of 5S rRNA (Fig. 7e, Supplementary Fig. 11), suggesting that ANG and its receptor PLXNB2 are responsible for the production of 5S rRNA fragments in CSCs.

**3′-end 5S rRNA fragment enhances stemness of CSCs.** To understand whether the 3′-end fragment of 5S rRNA regulates CSC stemness, we constructed plasmids expressing the 3′-end 5S rRNA fragments (47 nt), together with the 5′-end 5S rRNA fragment (47 nt), the full-length 5S rRNA (120 nt), and a scramble RNA (47 nt) as controls, transfected them into CSCs, and examined the resultant changes in stemness. Upon expression the 3′-end 5S rRNA fragment, PC3 CSCs displayed elevated expression of stemness-related genes (Fig. 8a), enhanced sphere formation (Fig. 8b), and reduced protein synthesis (Fig. 8c). Enhanced expression of stemness-related genes was also observed

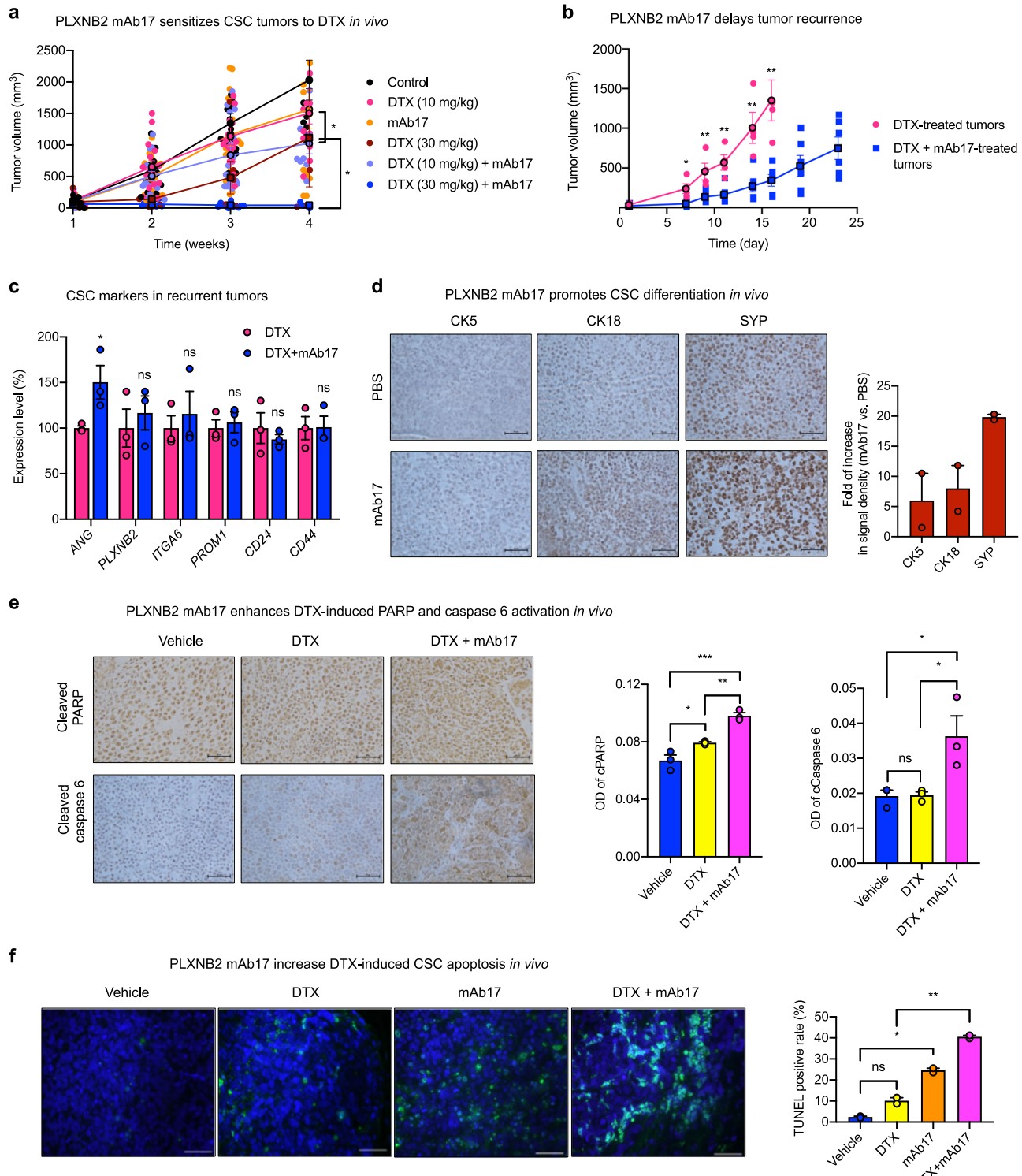

**Fig. 6 Effect of ANG and PLXNB2 mAbs on chemo-sensitivity of CSC tumors. a** PLXNB2 mAb sensitizes CSC tumors to DTX. NSG mice were inoculated with 100,000 CSCs, randomized into 6 groups ($n = 6$–19) when tumor sizes reached ~100 mm³, and treated with PBS control ($n = 6$), 4.8 mg/kg mAb17 ($n = 19$), 10 mg/kg DTX ($n = 16$), 30 mg/kg DTX ($n = 6$), 10 mg/kg DTX + 4.8 mg/kg mAb17 ($n = 16$), and 30 mg/kg DTX + 4.8 mg/kg mAb17 ($n = 10$), by weekly i.p. injection. Tumor sizes were measured weekly. **b** Treatment was ceased on week 4. Animals bearing similar tumor sizes in the group of 30 mg/kg DTX (4 mice with an average tumor size of 34.4 ± 5.6 mm³) and in the group of 30 mg/kg DTX + 4.8 mg/kg mAb17 (6 mice with an average tumor size of 22.2 ± 8.3 mm³) were observed for tumor growth for another 23 days. **c** mRNA levels of *ANG*, *PLXNB2*, and representative CSC markers in recurrent tumors determined by qRT-PCR ($n = 3$). Values were normalized to the average of the DTX group. **d** IHC staining of CK5, CK18, and SYP in CSC xenograft tumor tissues treated with PBS or mAb17 ($n = 2$). Scale bar: 50 μm. **e** IHC of cleaved PARP and cleaved caspase 6 in CSC xenograft tumor tissues treated with PBS, 30 mg/kg DTX, and 30 mg/kg DTX + 4.8 mg/kg mAb17 ($n = 3$). Scale bar: 50 μm. **f** Apoptotic cells detected by TUNEL assay in CSC xenograft tumor tissues treated with DTX (30 mg/kg), mAb17 (4.8 mg/kg), or both. Scale bar: 100 μm. TUNEL positive cells were counted in a total of 200 cells in each sample ($n = 2$). *$p < 0.05$; **$p < 0.01$; ***$p < 0.001$; ns not significant.

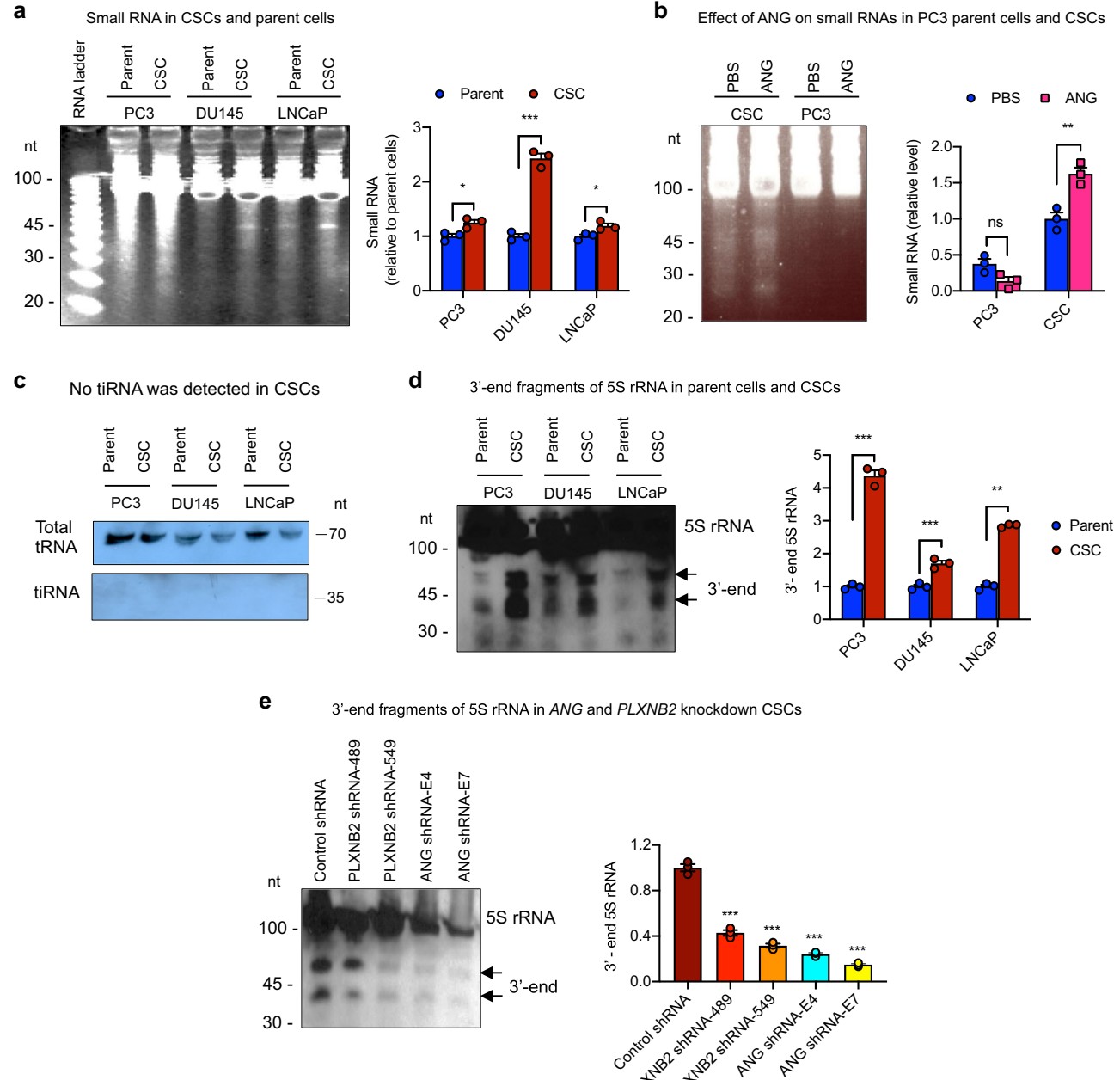

**Fig. 7 ANG mediates production of 3′-end fragment of 5S rRNA in prostate CSCs. a** Small RNA in CSCs and parent cells analyzed by polyacrylamide gel electrophoresis and SYBR Gold staining ($n = 3$). Band intensity was determined by Image J and presented as relative values normalized to respective parent cells. **b** Effect of ANG on small RNA production in CSCs and PC3 cells ($n = 3$). Band intensity was determined by Image J and presented as relative values normalized to PBS-treated CSCs. **c** Northern blot analysis of tRNA-Gly-GCC ($n = 2$). **d** Northern blot analysis of small RNA from prostate CSCs and respective parent cells using the 3′-end fragment of 5S rRNA as the probe ($n = 3$). Band intensity was determined by Image J and presented as relative values normalized to respective parent cells. **e** Level of the 3′-end fragment of 5S rRNA in *PLXNB2* and *ANG* knockdown CSCs analyzed by northern blotting, quantified by Image J, and normalized to control shRNA transfectants ($n = 3$). *$p < 0.05$; **$p < 0.01$; ***$p < 0.001$; ns not significant.

| Table 3 Identity and abundance of small RNA found in prostate CSCs. | |
|---|---|
| **Small RNA** | **Number of clones** |
| 5S RNA | 21 |
| tRNA-Asp$^{GAY}$ | 4 |
| tRNA-Asp$^{GGY}$ | 1 |
| tRNA-Asp$^{TGA}$ | 1 |
| U2 snRNA | 1 |
| Short sequences | 35 |

in CSCs of DU145 and LNCaP cell lines expressing the 3′-end 5S rRNA fragment (Fig. 8d). Consistently, we found that CSCs expressing the 3′-end fragment of 5S rRNA grew slower in NSG mice than that expressing the 5′-end fragment of 5S rRNA, the intact 5S rRNA, or the scramble control RNA (Fig. 8e). Tumors derived from CSCs expressing the 3′-end fragment of 5S rRNA were more resistant to DTX. While administration of 30 mg per kg DTX inhibited tumors derived from CSCs expressing a control RNA, the intact 5S rRNA, or the 5′-end fragment of 5S rRNA by ~50%, it inhibited tumors derived from CSCs expressing the 3′-end fragment of 5S rRNA by only 20% (Fig. 8e), indicating more

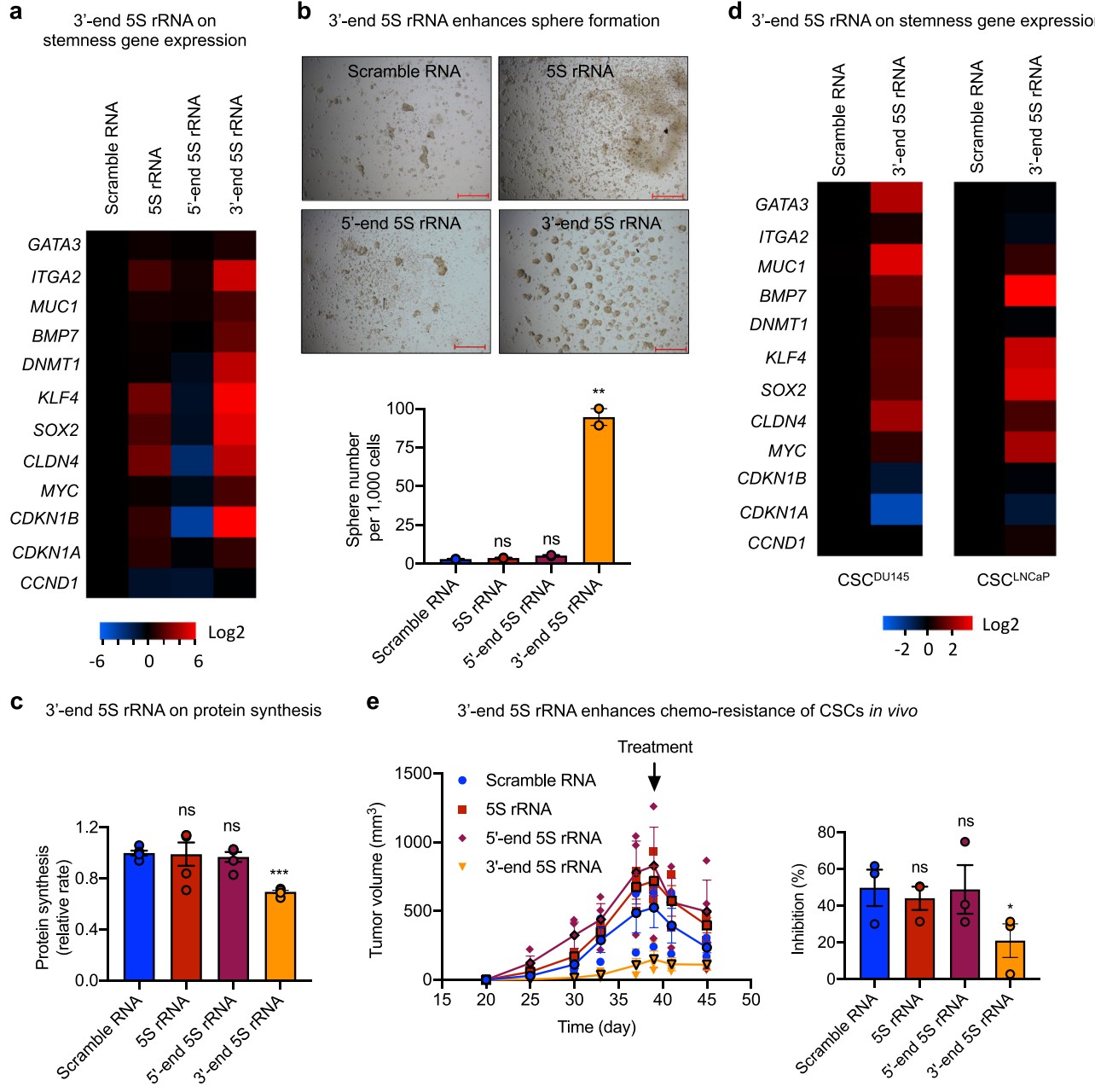

**Fig. 8 3′-end fragment of 5S rRNA enhances stemness of prostate CSCs. a** mRNA level of stemness-related genes in stable CSC transfectants of various small RNA ($n = 3$). Values were normalized to scramble RNA transfectants. **b** Sphere formation of stable CSC transfectants of various small RNA ($n = 2$). Scale bar: 500 μm. **c** Relative protein synthesis rate in stable transfectants of CSCs with various small RNA measured by OP-Puro incorporation followed by flow cytometry ($n = 3$). **d** mRNA level of stemness-related genes in stable transfectants of prostate CSCs of DU145 and LNCaP with scramble RNA control and the 3′-end fragment of 5S rRNA. **e** Xenograft growth of CSC transfectants of various small RNA. NSG mice were inoculated with 100,000 CSCs transfected with scramble RNA, intact 5S rRNA, the 5′- or 3′-end fragment of 5S rRNA, and treated with a single dose 30 mg/kg DTX (i.p.) on day 39. Bar graphs represent percent of inhibition of tumor growth in response to DTX treatment by comparing the tumor volume on day 45 to that on day 39 in each group. $*p < 0.05$; $**p < 0.01$; $***p < 0.001$; ns not significant.

chemo-resistant phenotype of CSC tumors overexpressing the 3′-end fragment of 5S rRNA.

In order to determine whether ANG also mediates the production of small RNA and whether the 3′-end fragment of 5S rRNA enhances stemness in other types of CSCs, we examined the effect of ANG in P19 embryonal carcinoma cells, a well-characterized cell line that has been shown to possess pluripotent CSC properties and to be able to differentiate into cell types of all three germ layers[48]. We found that ANG was able to enhance the production of total small RNAs in P19 cells in a dose-dependent

manner (Fig. 9a, Supplementary Fig. 13) as well as the production of the 3′-end 5S rRNA fragment (Fig. 9b, Supplementary Fig. 13). Transfection of the 3′-end 5S rRNA fragment in P19 cells resulted in a decrease in cell proliferation (Fig. 9c) and protein synthesis (Fig. 9d), an increase in sphere formation (Fig. 9e), enhanced expression of pro-survival protein Bcl-2, and decreased expression of Parp-1 and apoptosis inducing factor (AIF) (Fig. 9f, Supplementary Fig. 13). These results demonstrate that ANG is able to mediate the process of 5S rRNA and that one of the products, the 3′-end 5S rRNA fragment, is able to enhance the

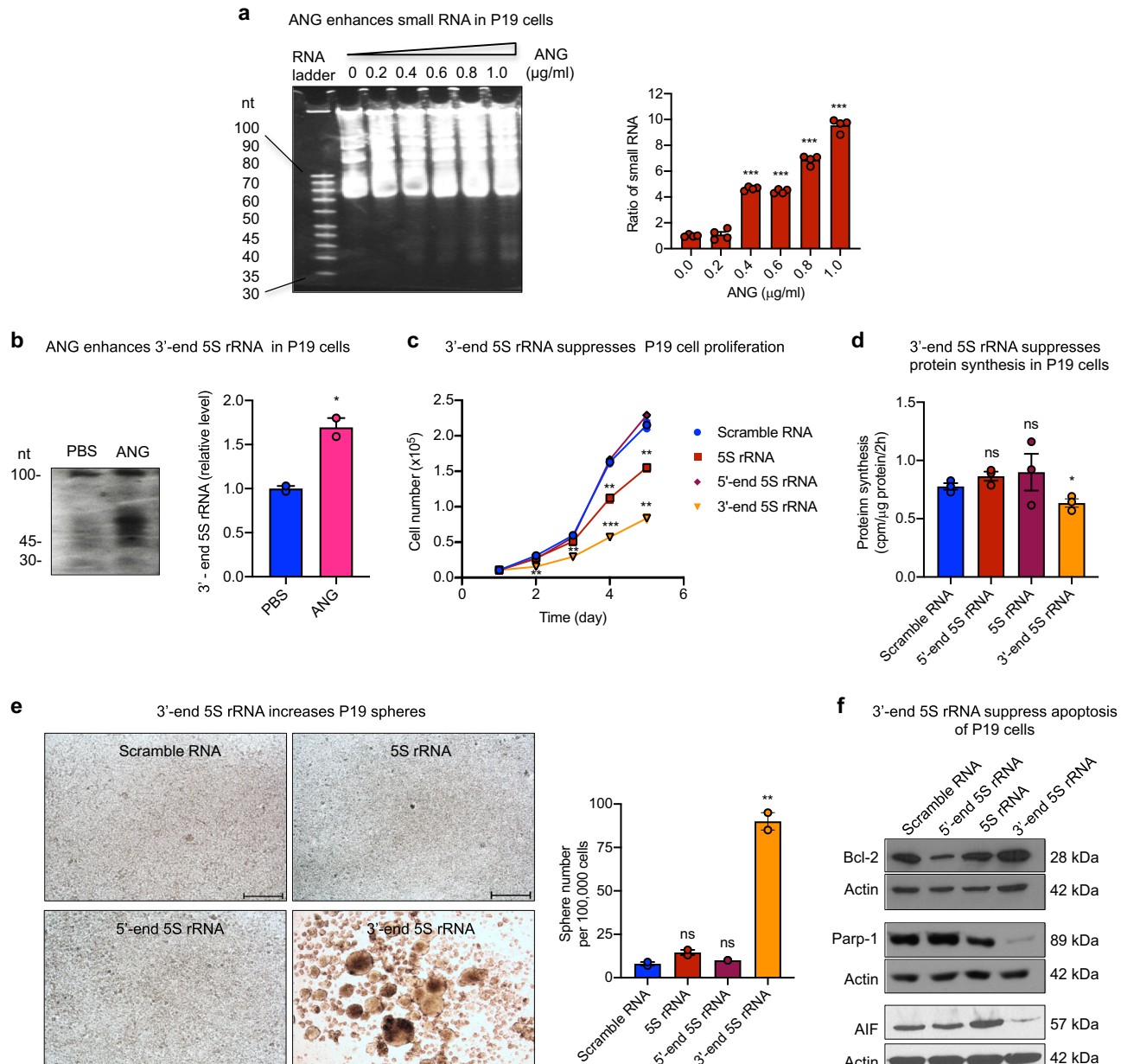

**Fig. 9 3′-end fragment of 5S rRNA enhances stemness of P19 cells. a** Small RNA in P19 cells treated with various concentration of ANG for 2 h. A total of 20 µg RNA was loaded on each lane. The intensity of small RNA bands was calculated by Image J and normalized to untreated control ($n = 4$). **b** Northern blot analysis of the 3′-end fragment of 5S rRNA in P19 cells treated with PBS or ANG (1 µg/ml) for 2 h. The intensity of the band around 45 nt was determined by Image J and normalized to PBS control. **c** In vitro proliferation of P19 cells transfected with scramble RNA, intact 5S rRNA, the 5′- or 3′-end fragment of 5S rRNA ($n = 2$). **d** Protein synthesis rate of stable P19 transfectants of various small RNA determined by [3][H] methionine incorporation ($n = 3$). **e** Sphere formation of stable P19 transfectants of various small RNA. Sphere numbers were counted from the entire well of 24-well plate containing 100,000 cells cultured in sphere-forming medium for 24 h ($n = 2$). Scale bar: 500 µm. **f** Immunoblots of Bcl-2, cleaved Parp-1, and AIF in stable P19 transfectants of various small RNA cultured in sphere-forming medium. Actin was used as loading control in each blot. *$p < 0.05$; **$p < 0.01$; ***$p < 0.001$; ns not significant.

stemness of P19 cells, suggesting that ANG-mediated production of the bioactive 3′-end fragment of 5S rRNA is able to regulate both human and mouse CSCs.

## Discussion

The significance of this study is multifold. First, prostate CSCs were isolated and shown to have self-renewal, differentiation, and tumor-initiating capacities, thus satisfying the three criteria for a true cancer stem cell[35]. The prostate CSCs cloned in this study

were highly quiescent, able to self-renew in the absence of serum, could differentiate into various lineages of prostate cells in vitro and in vivo, and were able to initiate xenograft tumors with a single cell inoculation. Isolation and validation of authentic CSCs not only enabled mechanistic studies of how cancer stemness is maintained and regulated, but also provided a much needed experimental model for therapeutic intervention of drug resistance and disease recurrence.

Second, we revealed a regulatory mechanism through which CSC stemness is maintained. We found that the inhibition of

ANG and PLXNB2 reduced CSC stemness and elicited CSC exhaustion. The dichotomous regulation of ANG-PLXNB2 towards prostate CSCs and differentiated bulk cancer cells is mechanistically unique. While ANG maintains quiescence and self-renewal of CSCs, it promotes proliferation of differentiated cancer cells, both functions are prominently related to cancer progression and metastasis.

We found that ANG is mainly located in stress granules of CSCs and in the nucleolus of differentiated prostate cancer cells. ANG was found to enhance the production of small RNAs in CSCs; however, we did not detect much tiRNA in[6]. Instead, we observed a large number of the 3′-end fragments of 5S rRNA in CSCs, which can be increased by exogenous ANG and diminished by knockdown of ANG or PLXNB2. Available data from this study indicate that it is the bioactive 3′-end fragment of 5S rRNA, but not the decrease of 5S rRNA, which promotes CSC stemness since transfection of the intact 5S rRNA actually slightly reduced CSC proliferation and decreased expression of stemness-related genes. Moreover, transfection of the 3′-end fragment of 5S rRNA greatly enhanced CSC stemness as reflected by higher G0 cell frequency, elevated expression of stemness-related genes, reduced protein synthesis, decreased cell proliferation, increased sphere formation, and enhanced resistance to chemotherapy. The molecular mechanism by which the 3′-end 5S rRNA fragment enhances CSC stemness will be a subject for future studies.

The third finding is that CSC stemness can be attenuated by ANG and PLXNB2 mAbs. The anti-prostate cancer activity of ANG mAb 26-2F has been well documented[17,49] and has served as proof-of-concept that inhibition of ANG activity would be an effective means for cancer therapy[8]. Recently, we identified PLXNB2 as the functional ANG receptor that is both necessary and sufficient to mediate physiological and pathological activities of ANG and have shown that PLXNB2 mAb prevented the establishment of PC3 xenograft tumors and inhibited the growth of established tumors[14]. In this study, both ANG and PLXNB2 mAbs, which were prepared in house from hybridoma cell culture and purified to homogeneity (>99% purity shown by SDS-PAGE) containing no preservatives or additives such as sodium azide and glycerol, were shown to reduce CSC stemness by enhancing their cycling as well as differentiation. Furthermore, we showed that ANG and PLXNB2 mAbs were able to sensitize CSCs to DTX treatment in vitro and in vivo. It is notable that PLXNB2 mAb was able to decrease the IC50 of DTX toward CSCs by fourfold, and that a combination therapy of PLXNB2 mAb and DTX had a synergistic effect to DTX-resistant CSC tumors and delayed disease recurrence upon cessation of treatment. Inhibitors of the ANG-PLXNB2 axis would be an effective therapeutic means for eliminating chemo-resistant CSCs that have been proven to hide in bone marrow niches[50] and causes drug resistance and disease relapse. Together with the robust antiangiogenic and anti-proliferative activities against differentiated bulk cancer cells, ANG-PLXNB2 inhibitors would thus hold promise for the treatment of castration-refractory, drug-resistant, and metastatic prostate cancers.

## Methods

**Animals**. NSG male mice (5–7 weeks of age) were purchased from the Jackson Laboratory and housed in sterile cages in the animal facility of Tufts Medical Center that is accredited by the Association for Assessment and Accreditation of Laboratory Animal Care, and visited daily by staff veterinarians. All animal experiments were approved by the Institutional Animal Care and Use Committee (IACUC) of Tufts University/Tufts Medical Center. All procedures were performed in accordance with protocols approved by the IACUC of Tufts University/Tufts Medical Center.

In the experiments to examine tumor-initiating capacity of CSCs, different doses of cells ranging from 1 to 5000 CSCs were prepared in 50 µl HBSS, based on the best estimation from serial dilutions, mixed with an equal volume of Matrigel

(Corning), and subcutaneously inoculated onto the flank region of the mice ($n = 8$–10). Tumors were examined 3 months post inoculation.

In the serial passaging experiments, 100,000 cells (CSCs stably transfected with control shRNA, ANG shRNA E4 and E7, PLXNB2 shRNA 489 and 549) in 50 µl HBSS were mixed with an equal volume of Matrigel and inoculated subcutaneously to the rear flank of NSG mice. At 4 weeks post inoculation, tumors were surgically removed, weighed, washed in sterilized HBSS, minced with repeated razor-cut into 3–4 mm pieces, and disaggregated in 1 ml of 0.25% trypsin (Hyclone, GE) with 15 min incubation at 37 ˚C. Tissue fibrosis was removed by straining cellular suspension through 0.45 µm meshes to obtain cell suspension. ACK Lysis Buffer (Sigma) was added to the cell pellet and incubated on ice for 5 min to deplete red blood cells. Cells were washed and resuspended in HBSS, counted using a hemocytometer after Trypan Blue staining for cell viability, and serially passaged in NSG mice three times.

In the experiment to examine chemo-sensitizing activity of PLXNB2 mAb, 10,000 CSCs per mouse was mixed with an equal volume of Matrigel and inoculated. Tumor appearance was examined by palpation. When the size of the tumors reached approximately 100 mm³, mice were randomized into 6 groups and treated with PBS control, 4.8 mg/kg mAb17, 10 mg/kg DTX, 30 mg/kg DTX, 10 mg/kg DTX + 4.8 mg/kg mAb17, and 30 mg/kg DTX + 4.8 mg/kg mAb17, by weekly i.p. injection. Tumor sizes were measured weekly and analyzed by one-way ANOVA followed by Newman–Keuls multiple comparisons test.

In the experiment to examine in vivo growth of CSCs transfected with scramble RNA, intact 5S rRNA, the 5′- or 3′-end fragment of 5S rRNA, 100,000 cells per mouse was mixed with an equal volume of Matrigel and inoculated. Treatment with 30 mg/kg DTX was giving by i.p. injection on day 39.

**Cells**. Human prostate cancer cell lines PC3 (male), DU145 (male), and LNCaP (male), mouse embryonal carcinoma cell line P19 (male) were purchased from ATCC. PC3, DU145, and P19 cells were cultured in DMEM (Sigma-Aldrich) + 10% FBS (Mediatech). LNCaP cells were cultured in RPMI1640 (Corning Cellgro) + 10% FBS. Human CD34+ cord blood cells (mixed genders, pooled sources) were purchased from Stem Cell Technologies and cultured in StemSpan SFEM (Stem Cell Technologies), supplemented with stem cell factor, Flt3 ligand, IL6, and thrombopoietin (100 ng/ml, R&D Systems). CSCs cloned from PC3, DU145, and LNCaP cells were maintained in sphere medium composed of DMEM: F12 (1:1) plus 20 ng/ml EGF, 20 ng/ml bFGF (Novus Biologicals) and 1x B27 Supplement (ThermoFisher Scientific) in bacteriological petri dish (Santa Cruz). All cells were cultured at 37 °C under humidified 5% CO2. Cell viability was determined by trypan blue exclusion method. Cells were screened for mycoplasma contamination every 3 months with e-Myco Mycoplasma PCR Detection Kit (Lilif, Cat # 25235). PCR-based short tandem repeat (STR) profiling of 9 markers (Abmgood, Cat # C287) was performed annually and compared with ATCC datasheet to authenticate the identity of PC3, DU145, and LNCaP cells.

**ANG and PLXNB2 knockdown**. The pLKO.1-puro lentiviral vector-shRNA system was used. To knockdown PLXNB2, shRNA TRCN00003-00489 (489) and −00549 (549) (Sigma Mission) were used. The sequences were: 489, 5′-CCGGGCTCTAC CAATACACGCAGAACTCGAGT TCTGCGTGTATTGGTAGAGCTTTTTG-3′; 549, 5′-CCGGGCAGAAGTACTATGACGAG ATCTCGAGATCTCGTCATAGT ACTTCTGCTTTTTG-3′. To knockdown ANG, shRNA lentiviral particles vectors (pGIPZ) E4 and E7 (Open Biosystems) were used. The sequences were: E4, 5′-ATGTTTGACAACATGTTTAATA-3′; E7, 5′-CAACGTTGTTGTTGCTTG TGAA −3′. A noncoding shRNA was used as the control. Lentiviral particles were prepared by transient transfection in HEK 293 cells using the ViraPower Lentiviral Expression Systems (Invitrogen) according to manufacturer's instruction. Lenti-viral particles were harvested after 72 h, centrifuged at 781 × g for 15 min, and filtered through a 0.45 µm PVDF membrane (Millipore). The viral particles were then ultracentrifuged at 83,000 × g for 1.5 h and the pellet was resuspended in PBS. pGIPZ lentiviral particles were packaged in HEK 293 cells with the generation II packaging plasmids (psPAX2 and pMD2.G) and concentrated by Lenti-X con-centrator (Clontech). Cells were infected with lentiviral particles for 24 h in the presence of Polybrene (8 µg/ml, Millipore). The medium was replaced with com-plete growth medium and incubated for 24 h and then selected for 4 days with puromycin at 6 µg/ml for CSCs and 0.5 µg/ml for parent cells and P19 cells.

**Annexin V apoptosis analysis**. Annexin V-PE Apoptosis Detection Kit (eBioscience) was used for apoptosis analysis following manufacturer's instruction. Briefly, cells were washed with ice-cold PBS, resuspended in 100 µl of 1x binding buffer at 1 × 10⁶ cells/ml, and incubated with 5 µl of PE-conjugated Annexin V for 15 min at RT in the dark. After washing with 1x binding buffer, 300 µl of 1x binding buffer was added to resuspend the cell pellet for analysis on a Cyan flow cytometer. Cell viability was determined by 7-AAD. Annexin V-positive gate was established by Annexin V fluorescence-minus-one controls.

**Array analysis**. GeneQuery™ human stem cell transcription factor qPCR array kit (ScienCell, Cat # GK125) was used per manufacturer's instruction. Total RNA from 100,000 cells was extracted by Trizol Reagent (Thermo Fisher Scientific) and reverse-transcribed to cDNA. For qPCR reaction, a mixture of 20 µl total volume

containing 2 μl of 20 ng/μl cDNA, 10 μl of SYBR Green PCR Mix, and 8 μl water was used for each well. The comparative ΔΔCq (quantification cycle value) method was used for quantification using *GAPDH* as control.

**Candidate CSC cloning by limited dilution**. Single cell cultures of PC3, DU145, and LNCaP cells were prepared by limited dilution in 96-well plate with a 100 μl per well regular culture medium. Any wells containing two or more cells were excluded from further study. The plates were maintained under normal culture conditions until the clones of single cell reached a size of ~200 μm² (~1.5 months). The clones with sphere-forming tendency (small cells, overlaying in clusters of round morphology, loosely attached) were detached by Versene and transferred to 35-mm bacteriological petri dish containing 1 ml sphere-forming medium (DMEM:F12, 1:1, containing 20 ng/ml EGF, 20 ng/ml bFGF, and 1× B27 supplement) and cultured for 4 months. Clones that survived and expanded under this culture condition were further characterized and examined for stemness. To passage the CSCs, the medium was removed by centrifugation, and the cell pellet was suspended in 100 μl of Accutase (Innovative Cell Technologies) and incubated at RT with frequent pipetting for 7 min, centrifuged, and resuspended in fresh sphere medium, split at 1 to 2 ratio, and cultured in bacteriological petri dishes.

**Cell cycle analysis**. Cells ($1 \times 10^7$) were fixed and permeabilized using Cytofix/Cytoperm Fixation/Permeabilization Kit (BD) and stained with anti-human Ki-67-PE (eBioscience, Cat # 12-5698-80, 1:20 in PBS + 2% FBS) and analyzed using a Cyan flow cytometer. DNA content was measured by 7-AAD.

**Cell transfection**. A mixture of 0.75 μg of target gene constructions and 0.5 μg of pBABE-puro (Addgene) plasmid was transfected into cells cultured in 24-well plate using Lipofectamine LTX DNA Transfection Reagents (Life technologies) per manufacturer's protocol. Puromycin (Life Technologies) was used for selection of stable transfectants at concentration of 6 μg/ml for CSCs and 0.5 μg/ml for PC3 and P19 cells. Stable transfectants of CSC, PC3, and P19 cells were maintained in medium containing 2, 0.2, and 0.5 μg/ml puromycin, respectively. For CSCs of DU145 and LNCaP, transient transfectants of plasmids expressing the scramble RNA control and the 3′-end fragment of 5S rRNA were harvested 72 h post transfection and used for experiments.

**Cloning and sequencing of small RNAs**. Small RNAs were enriched by Exact-START™ Small RNA Cloning Kit (Epicentre Biotechnologies) following manufacturer's instruction. Single-stranded small RNA was enriched from 200 μg total RNA, treated with alkaline phosphatase (Fermentas) followed by T4 polynucleotide kinase (Fermentas) to convert any possible 5′-hydroxyl and 3′-monophosphorylate ends to 5′-monophosphate and 3′-hydroxyl ends. After adding a poly(A) tail to the 3′-end and ligating a small RNA acceptor oligo containing a PCR priming site and a *Not*I restriction site to the 5′-monophosphate end, the modified single-strand small RNA was used as template for synthesis of first-strand cDNA with the small RNA cDNA synthesis primer, which consists of an oligo(dT) sequence, a PCR priming site and *Asc*I (Biolabs) restriction site at its 5′-end. PCR was performed to generate the second strand of cDNA and to amplify double-stranded cDNA for subsequent cloning. Double-stranded cDNA was digested with endonucleases *Asc*I and *Not*I, recovered by gel-purification, and ligated to the pre-cut pCDC1-K vector. The ligation mixture was then transformed to *E. coli* DH5α. Individual clones were selected and sequenced.

**Competitive engraftment analysis**. NSG mice were sub-lethally irradiated (2.5 Gy) 16 h prior to transplantation. Human CD34⁺ cord blood cells from mixed donors were washed once in PBS and 10,000 cells per mouse were intravenously injected into NSG recipients (*n* = 6). Both male and female mice were used as recipients. At 16 weeks post transplant, recipients NSG mice were sacrificed and bone marrow cells were collected. Red cell-depleted bone marrow mononuclear cells (BMMNCs) were stained with human CD45-pacific Blue (Biolegend, Ca # 304021) and analyzed by flow cytometry (FACSAria) to confirm successful engraftment of human CD34⁺ cells. BMMNCs from the first transplantation recipient mice were used as the donor cells for secondary transplantation experiments. A total of $5 \times 10^5$ BMMNCs per recipient was intravenously injected into 12 sub-lethally irradiated (2.5 Gy) NSG recipients. At week 2 post transplant, NSG recipients were randomly divided into two groups (six mice each) and injected intravenously with 500,000 of stable GFP-transfected CSCs or parent cells of PC3. After additional 4 weeks, red cell-depleted BMMNCs were surface stained with mouse CD45 APC-e780 (eBioscience, Cat # 47-0451-82) at a dilution of 1:200 on ice for 30 min in the dark. Samples were analyzed using a BD LSRII flow cytometer. GFP fluorescence signal was acquired to measure the engraftment of PC3 parent cells and CSCs.

**Construction of expression plasmids of 5S rRNA fragments**. pAAV-siRNA (Applied Viromics) vector was used for expression of 5S rRNA fragments. Double-stranded DNA was prepared by annealing two single-stranded oligonucleotide chains of complementary sequences. The sequences of the single-stranded DNA

were as follows. Scramble RNA control: forward 5′-tttGGCTCTAGTCTGCTCTAGCCCGCGCAAGTCCCACCATACCGGCATCTGtttt-3′, reverse 5′-gttacaaaa-CAGATGCCGGTATGGTGGGACTTGCGCGGGCTAGAGCAGACTAGAGC-3′. 5′-end fragment of 5S rRNA: forward 5′-tttGTCTACGGCCATACCACCCTGAAC GCGCCCG ATCTCGTCTGATCTCGGtttt-3′, reverse 5′-gttacaaaaCCGAGATC AGACGAGATCGGGCG CGTTCAGGGTGGTATGGCCGTAGA-3′. 3′-end fragment of 5S rRNA: forward 5′-tttGAGTACTTGGATGGGAGACCGCCT GGGAATACCGGGTGCTGTAGGCTTtttt-3, reverse 5′-gttacaaaaAAGCCTAC AGCACCCGGTATTCCCAGGCGGTCTCCCATCCAAGTA CT-3′. 5S rRNA: forward 5′-tttGTCTACGGCCATACCACCCTGAACGCGCCCGATCTCG TCTGATCTCGGAAGCTAAGCAGGGTCGGGCCTGGTTAGTACTTGGATG GGAGACCGCCTGGGAATACCGGGTGCTGTAGGCtttt-3′, reverse 5′-gtta-caaaaGCCTACAGCACCC GG TATTCCCAGGCGGTCTCCCATCCAAGTACT AACCAGGCCCGACCCTGCTTAGCTTCCGAGATCAGACGAGATCGGGCGC GTTCAGGGTGGTATGGCCGTAGA-3′. Two synthetic oligonucleotides with complementary sequences were annealed to form double-stranded DNA by mixing equal quantities of both oligos and heating at 95 ℃ for 2 min and then cooling down to RT. The double-stranded DNA was ligated to pAAV-siRNA vector pre-cut with *Bst*EII and *Bbs*I and transformed to *E. coli* DH5α. Sequences of inserted fragments were confirmed by DNA sequencing using U6 primer (5′-GAGACTA TAAATATCCCTTGGAG-3′) or a reverse primer (5′-AGAGAGGGAGTGGCCA ACT-3′).

**Flow cytometry analyses of stem cell markers**. Cells were detached by trypsin-EDTA, washed with ice-cold PBS, resuspended in PBS containing 2% FBS at $1 \times 10^6$ cells per 150 μl, and stained with anti-hCD133-PE (Miltenyi Biotec, Cat # 130-111-079) at dilution of 1:8, anti-hCD49f-PE (eBioscience, Cat # 12-0495-83) at 1:160, anti-hCD44-FITC (Miltenyi Biotec, Cat # 130-113-903) at 1:100, anti-hCD326 FITC (Bio-Rad, Cat # MCA1870F) at 1:100, on ice for 20 min in the dark. Stained cells were washed twice with PBS + 2% FBS at $350 \times g$ and suspended in 300 μl of PBS + 2% FBS for analysis on a Cyan flow cytometer. For CD24 and Trop2, unconjugated primary antibodies (eBioscience, Cat # 14-0247-82 and 14-6024-82;) were used at 1:100 dilution, followed by PE conjugate of goat F(ab')₂ anti-mouse IgG (eBioscience, Cat # 12-4010-82) at dilution of 1:125. PI or 7-AAD was used to exclude dead cells. For ALDH analysis, ALDEFLUOR™ kit (Stemcell Technologies) was used. Briefly, 1 ml of $1 \times 10^6$ cells was mixed with 5 μl of activated ALDEFLUOR™ reagent, half of the cells (0.5 ml) was immediately pipetted to control tube containing 5 μl of DEAB reagent, and both tubes were then incubated in a cell incubator for 45 min. Cells were then washed with ALDEFLUOR™ buffer and centrifuged at $350 \times g$ for 5 min. Cell pellets were suspended in 0.5 ml of ALDEFLUOR™ buffer for analysis on a Cyan flow cytometer.

**Immunofluorescence**. Cells were trypsinized and cultured on cover slips placed in 48-well plates in PBS containing 10% BSA overnight. Cells were fixed in methanol at −20 ℃ for 10 min, blocked with 30 mg/ml BSA at 37 ℃ for 30 min, and incubated with the flowing antibodies, a mouse anti-human CK18 mAb (DC10, Cell Signaling, Cat # 4548) at 1:500 dilution, rabbit anti-human CK5 mAb (EP1601Y, Novus Biologicals, Cat # NB110-56916) at 1:200, rabbit anti-human SYP mAb (SP15, Enzo Life Science, Cat # ADI-VAM-SV011-F) at 1:400, rabbit anti-human ANG polyclonal IgG R113 (made-in-house) at 10 μg/ml, and mouse anti-PABP mAb (10E10, Abcam, Cat # ab6125) at 6 μg/ml in PBS containing 5% BSA at 4 ℃ overnight. After washing with PBS containing 5% BSA three times, 10 min each time, cells were incubated with goat F(ab')₂ anti-mouse IgG-Alexa Fluor 555 (Invitrogen, Cat # A21425) or goat F(ab')₂ anti-rabbit IgG-Alexa Fluor 488 (Invitrogen, Cat# A-11070) accordingly at 1:400 at 37 ℃ for 1 h. The cover slips were mounted on glass slides with Prolong Gold Antifade Reagent with DAPI (Molecular Probes) and fluorescent images were taken on a Nikon eclipse Ti fluorescent microscope.

**Immunohistochemistry**. Formalin-fixed, paraffin-embedded tumor sections were dewaxed and rehydrated by heating at 60 ℃ for 40 min followed by incubating for 5 min each in 100, 95, 90, 80, and 70% xylene in ethanol. After washing in tap water, slides were soaked completely in a beaker half filled with boiling sodium citrate buffer (10 mM sodium citrate, 0.05% Tween 20, pH 6.0) and boiled for 20 min in a microwave to retrieve the epitopes. Slides were washed with TBS containing 0.025% Triton X-100 twice, blocked in 10% goat normal serum with 1% BSA in TBS at RT for 2 h, and incubated with primary antibodies diluted in TBS containing 1% BSA in a humidified incubator at 4 ℃ in the dark overnight. The antibodies against BCL-2 (Cat # 15071), p53 (Cat# 9282), cleaved caspase 6 (Cat# 9761), cleaved p-PARP (Cat # 9548) were from Cell Signaling Technology and were used at 1:400 dilution. The polyclonal antibody against Ki-67 (EMD Millipore, Cat # AB9260) was used at 1:300 dilution, anti-PLXNB2 mAb17 (mad-in-house) was used at 5 μg/ml. Endogenous enzymes were blocked with 0.3% hydrogen peroxide in TBS for 15 min at RT. Goat anti-rabbit IgG-HRP (Bio-Rad, Cat # 1706515) or goat anti-mouse IgG-HRP (Bio-Rad, Cat # 1706516) were applied to the slides at 1:500 dilution in TBS containing 1% BSA and incubated for 1 h at RT. Signals were developed using Peroxidase Substrate Kit DAB (Vector Laboratories, Inc.) per manufacturer's instruction. The specificity of primary antibodies was confirmed by immunoblotting. The specificity of secondary antibody was confirmed by replacing

primary antibody with an isotype IgG control. Hematoxylin was used to counterstain the nuclei. Slides were dehydrated and mounted with coverslips using SP15-500 Toluene Solution (Fisher). Images were taken by a Nikon eclipse Ti microscope.

**Northern blot analysis**. Total RNA was isolated and separated by denaturing urea polyacrylamide gel electrophoresis and transferred to a Pall Biodyne nylon membrane (Promega) in pre-chilled 0.5X TBE at 80 V for 60 min at 4 °C. Transfer efficiency was confirmed by post transfer staining of the gel with SYBR Gold. The membrane was rinsed in 2X SSC buffer and fixed by baking at 248 °F for 30 min, and stored at RT between clear protection sheets. Before blotting, the membrane was rinsed in pre-warmed digoxigenin (DIG) Easy Hyb buffer (Roche) for 30 min at 50 °C with rotation and then hybridized in DIG Easy Hyb buffer containing heat-denatured DIG-labeled DNA Probe (IDT) at 25 ng/ml. The sequences of the HPLC-purified DIG-labeled probes are as follows. 5′-tiRNA Gly-GCC: 5′-GGCG AGAATTCTACCACTGAACCACCAA-3′, 3′-end fragment of 5S rRNA: 5′-AAG CCTACAGCACCCGGTATTCCCAGG-3′, 5′-end fragment of 5S rRNA: 5′-TCG GCGCGTTCAGGGTGGTATGGCCG-3′. After hybridization overnight, membranes were rinsed once in 2X SSC + 0.1% SDS for 10 min at 60 °C, twice in 0.5X SSC + 0.1% SDS for 20 min at 60 °C and once for 5 min in Washing Buffer (Roche) at RT in a hybridization oven (VWR Scientific). The membranes were then blocked in blocking solution (Roche) for 30 min at RT and probed with alkaline phosphatase-labeled anti-DIG IgG (Roche, Cat # 11585762001) at dilution of 1:10,000 for 30 mins at RT, washed twice for 20 min per wash with Washing buffer (Roche), equilibrated for 5 min in detection buffer, incubated with substrate CSPD (Roche) for 10 min at 37 °C, and exposed to autoradiography film (Labscientific, Inc.).

**Protein synthesis by [³H] methionine incorporation**. P19 cells were cultured in 24-well plate in DMEM + 10% FBS, washed with methionine/cysteine-free DMEM, and supplied with freshly prepared medium containing [³H] methionine at a final concentration of 0.23 mCi/μmol in methionine/cysteine-free DMEM for 2 h. At the end of incubation, cells were washed twice in PBS containing 1 mg/ml of unlabeled methionine and lysed in 0.4 ml of RIPA buffer (20 mM sodium phosphate, 150 mM NaCl, 5 mM sodium pyrophosphate, 5 mM EDTA, 1% Triton X-100, 0.5% sodium deoxycholate, 1 mM sodium orthovanadate, 0.1% SDS, pH 7.4) supplemented with 1 tablet/10 ml of complete mini-EDTA protease inhibitor cocktail (Roche) and 1% BSA on ice for 20 min. The lysate was centrifuged at $12,000 \times g$ for 15 min, an aliquot of 10 μl of the supernatant was taken for measurement of protein concentration by Bradford assay, the remaining lysate was precipitated by TCA at a final concentration of 10% at 4 °C for 10 min. Protein pellet was collected by centrifugation for 30 min at $12,000 \times g$ at 4 °C, resuspended in 50 μl of 0.2 M NaOH, and measured for radioactivity by liquid scintillation counting.

**Protein synthesis analyses by OP-Puro incorporation**. Protein synthesis rates in prostate CSCs and parent cells were measured using OP-Puro incorporation followed by flow cytometry as previously described[6]. Briefly, cells were cultured for 1 h with OP-Puro (Medchem Source) at a final concentration of 50 μM in routine medium with or without 30 μg/ml of mAbs at 37 °C. At the end of incubation, cells (both CSC spheres and attached parent cells) were trypsinized and washed once with $Ca^{2+}$- and $Mg^{2+}$-free PBS, fixed with 1% paraformaldehyde (Affymetrix) on ice for 15 min, washed once with PBS, and then permeabilized with 3% FBS and 0.1% saponin (Sigma) for 5 m at RT. Click-iT Cell Reaction Buffer Kit (Life Technologies) was used for azide-alkyne cycloaddition of AF488-conjugated azide (5 μM, Life Technologies) following manufacturer's instructions. Cells were washed twice in PBS + 2% FBS, resuspended in 300 μl of PBS + 2% FBS, and analyzed on a Cyan flow cytometer. FlowJo was used to calculate Geometric Means.

**PSA measurement**. CSCs and parent cells were cultured in DMEM + 10% FBS in 35-mm dishes at a seeding density of 50,000 per dish for 24 h. Culture medium was collected and the amount of PSA was determined by ELISA (MyBioSource). The average of the starting and ending cell numbers was used for calculation of PSA secretion from each cell.

**qRT-PCR analyses**. Total RNA was extracted using TRIzol RNA reagent (ThermoFisher). Total RNA (1 μg) was reverse-transcribed in a 25 μl volume system containing M-MLV reverse transcriptase (Promega) and Oligo dT primer or random primers (IDT). qRT-PCR analysis was performed on a LightCycler 480 II (Roche) using SYBR Green PCR mix (Roche). Relative expression was determined by the $2^{-\Delta\Delta Ct}$ method, and heat-map was plotted using $\log_2$ (fold of change) by RStudio (http://www.rstudio.com). GAPDH and glucose-6-phosphate isomerase (GPI) were used as housekeeping gene controls. See Supplementary Table 2 for primer information.

**Small RNA gel electrophoresis**. Total RNA was extracted by TRIzol and quantified by NanoDrop (Thermo Scientific). A total of 20 μg of RNA was diluted in Novex TBE-Urea sample buffer (Invitrogen), heated at 65 °C for 5 min, and cooled to RT prior to loading. A low molecular weight marker ladder (10–100 nt,

Affymetrix) was heated and cooled at the same time. A 15% TBE-Urea Gel (Invitrogen) was pre-run at 74 V for 60 min and samples were loaded and electrophoresed to the bottom of the gel at 100 V in 1x TBE running buffer. The gel was stained with SYBR Gold solution (Invitrogen) for 60 min with agitation and imaged on a Kodak Electrophoresis Documentation and Analysis System 120 with UV illumination.

**Soft agar colony formation**. Cells (1250) were mixed in 0.5 ml of 0.3% agarose in sphere medium and seeded onto 24-well plates pre-coated with 0.5% agar. Plates were incubated for 2 h at 37 °C, added with 0.25 ml of fresh medium per well, and incubated for 15 days with the addition of 0.25 ml fresh medium every 4 days. The number of colonies was counted using a dissecting microscope at the end of the experiment. Pictures of colonies were taken by using a Nikon Eclipse Ti microscope.

**Spheroid invasion**. Invasion of PC3 and CSCs was examined by a hanging-drop method as described[46]. Briefly, single cell suspension (50,000 cells in 30 μl sphere medium) was cultured on the lid of a 100-mm culture dish for 2 days. Spheroids were collected and mixed with pre-cold culture medium (DMEM + 10% FBS) containing 1.5 mg/ml collagen (BD) and 0.15 mg/ml laminin (Sigma). While the gel was still in liquid phase, 50 μl of mixture containing single spheroid was transferred into 96-well plate. After gel was solidified, 100 μl of complete medium was added. For quantification, the number of active invasive cells was counted in an area with a radius that was 1.5 times of the spheroid. The total invasion area was measured by Image J by connecting the farthest invaded cells.

**Tumor sphere formation**. Prostate CSCs or parent cells (1000–10,000) were seeded in 2 ml sphere-forming medium in 35 mm bacteriological petri dish and cultured for 7 days. P19 cells (100,000 cells per well of 24-well plate) were seeded in 0.5 ml sphere-forming medium for 24 h. The number of tumor spheres was counted using a dissecting microscope at the end of the experiment. Images of spheres were taken by using a Nikon Eclipse Ti microscope.

**TUNEL assay**. Apoptosis rate in tumor tissues was determined by In Situ Cell Death Detection Kit (Roche) per manufacturer's instruction. Formalin-fixed, paraffin-embedded tumor sections were dewaxed and rehydrated by heating at 60 °C for 40 min followed by incubating for 5 min each in 100, 95, 90, 80, and 70% xylene in ethanol, rinsed in PBS, permeabilized in freshly prepared 0.1% Triton X-100, 0.1% sodium citrate solution for 8 min, and rinsed again with PBS. TUNEL reaction mixture (50 μl) was added and incubated for 60 min at 37 °C in a humidified chamber in dark. A negative control was set up simultaneously and processed with the same procedures. The slides were rinsed three times with PBS and mounted with cover glasses using antifade mounting solution, and analyzed under a Nikon eclipse Ti fluorescence microscope. To detect apoptosis rate in cells, $2 \times 10^6$ cells in 100 μl PBS were mixed with 100 μl of 4% PFA at RT for 60 min, washed once with PBS, permeabilized in freshly prepared 0.1% Triton X-100, 0.1% sodium citrate solution for 2 min on ice, washed twice with 200 μl of PBS, resuspended in 50 μl of TUNEL reaction mixture and incubated for 60 min at 37 °C in a humidified chamber in the dark.

**Western blot analysis**. A total of 20 μg protein was separated by SDS-PAGE and electro-transferred to a nitrocellulose membrane in Towbin system buffer. The membrane was blocked with 5% fat-free milk in TBST and incubated with the primary antibodies at 4 °C overnight followed by incubation with the proper secondary antibodies (either goat anti-rabbit IgG-HRP or goat anti-mouse IgG-HRP) (Bio-Rad) at RT for 1 h. The following antibodies were used, ANG polyclonal antibody R113 (made-in-house) at 2 μg/ml, PLXNB2 mAb mAb17 (made-in-house) at 2 μg/ml, AIF polyclonal antibody (Cell Signaling, Cat # 4602) at 1:1000, cleaved PARP-1 rabbit mAb (Cell Signaling, Cat # 5625) at 1:1000, and Bcl-2 rabbit mAb (Cell Signaling, Cat # 3498) at 1:1000, and β-actin mAb (Santa Cruz, Cat # sc-47778) at 1:600 dilution.

**Statistics and reproducibility**. For immunoblots, Image J (NIH) was used to quantify band intensities. Multiple independent experiments were normalized and averaged, the average and standard deviation was presented. Biological and technical replications were indicated by a number *n*. For qRT-PCR analysis, the $2^{-\Delta\Delta Ct}$ method was used to quantitate relative expression, with GPI or β-actin as an internal control. Error bars represent standard deviation. All heatmaps (RStudio, http://www.rstudio.com) represent mean from 2–4 independent experiments, unless otherwise indicated. Unpaired two-tailed Student's *t* test (Prism 7) was used for comparison of two experimental groups. A one-way ANOVA followed by Newman–Keuls multiple comparisons test was used for comparison of multiple groups (SPSS v20). No specific inclusion and exclusion criteria were used for any data or subjects. No methods were used to determine whether the data met assumptions of the statistical approach. For all analyses, $*P < 0.05$; $**P < 0.01$; $***P < 0.001$; ns not significant.

**Reporting summary**. Further information on research design is available in the Nature Research Reporting Summary linked to this article.

## Data availability

No datasets were generated or analyzed during the current study. All data generated or analyzed during this study are included in this published article. Source data underlying plots are provided in Supplementary Data 1–10. Full blots and gels are shown in Supplementary Information.

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

## Acknowledgements

This research was supported in part by NIH grants R01CA105241 and R01HL135160 (to G-F. Hu), NSFC Grant #81272674 and Natalie V. Zucker Award (to S. Li).

## Author contributions

S.L. and G.F.H. conceived the project, designed experiments, and analyzed data. S.L., K.A.G., B.L. L.Y. performed experiments. G.F.H. supervised the project. S.L. and G.F.H. wrote the manuscript.

## Competing interests

G.F.H. is a non-paid consultant of Karma Pharmaceuticals Limited. All the other authors have no competing interests.
