## [Peer Review File · Communications Biology]

Reviewers' comments:

Reviewer #1 (Remarks to the Author):

The paper by Shuping Li et al. entitled "Chemosensitization of prostate cancer stem cells by angiogenin and plexin-B2 inhibitors" describes a series of experiments that characterize derivatives of a few established prostate cancer cell lines PC3, DU145 and LNCaP that were maintained in sphere forming medium. Specifically, these cells have increased sphere forming abilities, tumor initiating capacities and resistance to chemotherapeutic drugs. The authors demonstrated that prostate cancer cells cultured in sphere forming conditions are regulated by ANG and PLXNB2 proteins through the biogenesis of 3'-end 5S rRNA fragments. Overall, this study can be of general interest to prostate cancer researchers; however there are a few concerns.

1. Materials and Methods section:

- The description of the cell lines used for the study is missing; the authors need to mention if cell line authentication and mycoplasma testing were performed prior the experiments;
- the indicated numbers of cells used for sphere forming assay are too high (10,000 or 100,000 cells per 2 ml). Were the formed structures protospheres or the cells aggregates?
- The multiple generations of the spheres are described neither in text nor in the Materials and Methods section.

2. Page 4: tumor take rare was analyzed by inoculation of the sphere-derived cells. Is there significant difference in CSC frequency as compared to the parental cells?

3. Did the authors check the CSC phenotype of the sphere-derived cells e.g. ALDH(hi)CD44(+) α 2 β 1(+) cell population, PSA expression in LNCaP cells etc. To make the analysis of differentiation on Figure 2A more accurate and unbiased, a simple array should be performed to see how SC-related genes are regulated in the parental and sphere-derived cells.

4. How reliable are basal and luminal markers analysis used in the study to determine the differentiated phenotype? The previous studies showed that CRPC CSC ALDH+/CD44+/ α 2 β 1+ cells have a high expression of basal cell genes (e.g. CK5, CK14) whereas these markers were not detected in LNCaP cells (Tang DG et al, Clin Cancer Res 2016; Oncotarget 2015).

5. The authors are encouraged to characterize a potential correlation of ANG and PLXB2 expression with patients' clinical outcome in the publicly available prostate cancer gene expression data sets.

6. Did the authors use Isotype IgG as control treatment for the experiment on Figure 4 A?

7. How did the authors control for the specificity of the IHC staining?

8. The authors need to report core information about the antibodies that were used in their studies including the antibody name, the company, the catalogue/clone numbers.

9. The manuscripts would substantially benefit from a careful proofreading.

Reviewer #2 (Remarks to the Author):

The manuscript by Hu and colleagues addressed the question how prostate cancer stem cells (CSC) survive chemotherapy. They discovered a novel mechanism how angiotensin is regulating prostate CSC survival via plexin-B2 and 5S RNA. Within a previous publication from same group of Hu they discovered the connection how the ANG-Plexin-B2 pathway is regulation cell growth and proliferation via the AKT/ERK pathway, rRNA transcription and ribosomal biogenesis and via tRNAs under physiological and pathophysiological conditions (Yu et al. 2017, Cell). Other groups demonstrated that in prostate cancer the oncogene c-myc is regulating cell cycle arrest, differentiation and apoptosis via the 5S RNP complex and mdm2/p53. The role of the ANG-PLXNB2-5S RNA for prostate CSC survival is not known so far and is in detail analyzed within the submitted manuscript. Despite the manuscript is well written, easy to follow and the figures are illustrative, there are some points, which should be

addressed before publication in Communications Biology.

1. Within the introduction you are stating on page 4 that no reliable prostate CSC marker exist. This is not true, despite I agree that from most of the used marker functional testing is missing. But established markers, such as CD44+, CD133+, integrins, ALDH1A1, ...and should be at least mentioned.
2. Regarding the single cell experiments, please indicate how many clones were analyzed, what was the clonal distribution and what the clonal heterogeneity. Did you observe differences within the 3 used cell lines PC3, DU145 and LNCaP? (Fig. 1b)
3. In Fig. 1d the self-renewal was analyzed from the isolated single cell clones, despite the sphere-forming capacity is twice increased compared to the parental cell line. The clones loose the self-renewal potential during in vitro culture. A critical discussion would be necessary. I would also suggest including limiting dilution and serial passaging analysis here.
4. In Fig. 1e the author's shows that, the selected clones are mainly in G0 phase. Beside flow cytometry analysis, cell growth assays are necessary here. Are these cells still metabolic active? Are they senescent? Do you see differences in your 3D sphere-forming culture compared to the adherent culture? What is the potential of these quiescent cells to reactive proliferation and which rate?
5. In Fig. 1g the authors demonstrate bone tropism of the selected clones. Did you analyze any other organ for metastasis, e.g. lymph nodes?
6. For the whole manuscript the authors use CK5, CK14, CK18 and SYP/CHGA as surrogate marker for prostate CSCs. The used marker are exclusively labelling cells from the three described cell lineages within the prostate: basal, luminal and endocrine, but cannot be used to determine CSC count. Please add described markers, such as CD133, Aldefluor (ALDH1A1), Integrins, CXCR4, Nanog, Sox2, b-catenin, ... and EMT markers (in Fig. 3a).
7. In fig 2e and 2f the authors stating in the text that tumor growth pattern and histology is different in clones vs. parental cell lines. I do not see this in the figure with the used markers. Is the tumor stage different (e.g. using HE staining)?
8. Because Plexin-B2 is a cell membrane receptor, I was wondering if it would be possible to isolate the positive cells via FACS and compare to the negative cells concerning cell proliferation, sphere-formation, chemotherapy response and tumorigenicity.
9. To demonstrate the role of the ANG-Plexin pathway for cell migration, cell migration and invasion assays would be necessary upon ANG capturing.
10. What are the effects of the therapeutic antibodies 26-2F and mAb17 on normal tissue? Did you observe any late effect toxicities?
11. Fig. 6 is demonstrating delayed tumor growth upon DTX or mAb17 treatment, beside the CK IHC staining it would be necessary to analyze clonal composition and CSC marker expression in the recurrent tumors.

Minor comment:

1. Include calculation for synergy in Fig. 6a.
2. The IHC images are too small. Please include also images with higher magnification. (Fig. 2e,f, Fig.6c,d)

Reviewer #3 (Remarks to the Author):

It is very interesting that angiogenin-plexin-B2 (ANG-PLXNB2) pathway plays a critical role in the regulation of cancer stemness of prostate cancer. Furthermore, the possibility that the inhibition of the ANG-PLXNB2 pathway can conquer the chemoresistance in the prostate cancer could be benefit for future cancer treatment. Thus, this paper will be worthy to be published, but there are a few things to be modified for more confirmation.

1. Although the findings are novel, but it's very curious how the authors happened to find the involvement of the ANG-PLXB2 pathway in the regulation of cancer stemness. There is no description about it.
2. As you know that antibodies mostly obtained commercially have some additives, then those additives often interfere with cell growth. Thus, it is important to use control antibodies such as normal IgG. It is recognized that some experiments were done using such control antibodies, but not all. At least Fig. 4a, Fig. 5g, Fig. 6 should have control antibodies.
3. It is surprising that PC-3 and DU-145, androgen-independent cancer cells, potentially have AR expression abilities. It needs some reference papers.

We thank the reviewers for their insightful criticisms, comments and suggestions. We have performed experiments to address the issues raised by the reviewers. New figures (Fig. 1g, 1h; Fig. 1d, middle and right panels; Fig. 2e and 2f, middle panels; Supplementary Fig. 1a-1b, 4a-4g, 6a-6f, 8a-8d) are added, and the manuscript has been substantially revised.

The following is the point-by-point response to the criticisms, questions, and comments of the reviewers.

Reviewer #1

The paper by Shuping Li et al. entitled “Chemosensitization of prostate cancer stem cells by angiogenin and plexin-B2 inhibitors” describes a series of experiments that characterize derivatives of a few established prostate cancer cell lines PC3, DU145 and LNCaP that were maintained in sphere forming medium. Specifically, these cells have increased sphere forming abilities, tumor initiating capacities and resistance to chemotherapeutic drugs. The authors demonstrated that prostate cancer cells cultured in sphere forming conditions are regulated by ANG and PLXNB2 proteins through the biogenesis of 3'-end 5S rRNA fragments. Overall, this study can be of general interest to prostate cancer researchers; however there are a few concerns.

Response: We thank the reviewer for the positive comments and enthusiasm.

1. Materials and Methods section:

- The description of the cell lines used for the study is missing; the authors need to mention if cell line authentication and mycoplasma testing were performed prior the experiments;

Response: The source of cell lines, as well as the authentication and mycoplasma testing are described in the revised manuscript on page 22, lines 17-21 that “Cells were screened for mycoplasma contamination every 3 months with e-Myco Mycoplasma PCR Detection Kit (Lilif, Cat # 25235). PCR-based short tandem repeat (STR) profiling of 9 markers (Abmgood, Cat # C287) was performed annually and compared to ATCC datasheet to authenticate the identity of PC3, DU145, and LNCaP cells”.

- the indicated numbers of cells used for sphere forming assay are too high (10,000 or 100,000 cells per 2 ml). Were the formed structures protatospheres or the cells aggregates?

Response: We used 10,000 prostate cells (both parent cells and CSCs) in 35-mm dishes with 2 ml medium for sphere formation, which corresponds to 5,000 cell per well per 0.1 ml medium if it were done in 96-well plates. One of the reasons that a seemingly high number of cells was used in large dishes was that we aimed to compare sphere formation between CSCs and parent cells. For parent cells to form appreciable number of spheres, 10,000 cells were needed. For the same reason, we used 100,000 P19 cells in 25-mm dishes in 2 ml so that enough spheres could form for us to test the effect of small RNAs. We have now performed limited dilution assay with 100, 500, 1,000, and 5,000 cells and included the results in Fig. 1d, middle panel and on page 6, lines 3-4 where we stated that “Similar results were obtained in limited dilution analysis with PC3 CSCs (Fig. 1d, middle panel)”.

Prostaospheres and cell aggregates were distinguished by their distinct morphologies, as described on page 5, lines 2-4 from bottom that “The prostatospheres were identified morphologically as structures with clear membrane-like circle boundaries and were differentiated from cell aggregates that displayed a polymorphic structure”.

- The multiple generations of the spheres are described neither in text nor in the Materials and Methods section.

Response: we have examined sphere formation for 5 consecutive generations and found no appreciable decrease in sphere-forming ability. These new data were included as Fig. 1d, right panel, and described on page 6, lines 4-6 that “No appreciable decrease in sphere-forming ability of PC3 CSCs was noted for at least 5 passages in serial re-plating experiments in sphere medium in non-adherent dishes (Fig. 1d, right panel)”.

2. Page 4: tumor take rare was analyzed by inoculation of the sphere-derived cells. Is there significant difference in CSC frequency as compared to the parental cells?

Response: Yes, we described on page 5, lines 15-17 that “Under the same conditions, at least 50,000 parent PC3 cells were required to initiate xenograft tumors in NSG mice (one of six mice)”.

3. Did the authors check the CSC phenotype of the sphere-derived cells e.g. ALDH(hi)CD44(+) $\alpha 2\beta 1(+)$ cell population, PSA expression in LNCaP cells etc. To make the analysis of differentiation on Figure 2A more accurate and unbiased, a simple array should be performed to see how SC-related genes are regulated in the parental and sphere-derived cells.

Response: We discussed the heterogeneous phenotypes of CSCs based on the expression of basal, luminal, and neuroendocrine markers on page 8, lines 3-7, where we state that “The observation of an universal decrease in CK5, CK14, and CK18 expression in the three CSC lines as compared to their corresponding parent cells is in contrast to a previous report that CSCs with the phenotype of ALDH⁺CD44⁺ $\alpha 2\beta 1^+$ had high expression of CK5 and CK14³⁸, suggesting the heterogeneous nature of the CSCs”. We have also described the findings on intermediate cell markers and neuroendocrine markers on page 8, line 7-13 that “no consistent expression patterns of the intermediate cell markers were noted as CK19 was decreased in CSCs of PC3 but enhanced in CSCs of DU145 and LNCaP, whereas glutathione-S-transferase-pi (GSTpi) was decreased in CSCs of DU145 but increased in CSCs of PC3 and LNCaP. However, at least one of the two neuroendocrine markers was elevated in all three CSC lines: synaptophysin (SYP) was enhanced in CSCs of PC3, chromogranin A (CHGA) was elevated in CSCs of DU145, and both SYP and CHGA were elevated in CSCs of LNCaP (Fig. 2a and Supplementary Fig. 2)”.

We have measured PSA levels of CSCs and parent cells of LNCaP and the results were described on page 9, lines 5-8 from bottom that “A decrease in PSA level has been reported as an indicator of stemness of prostate cancer cells⁴³. However, no significant difference in PSA levels was found between CSCs (0.62 ± 0.09 ng/ 10^3 cells/day) and parent cells (0.65 ± 0.07 ng/ 10^3 cells/day) of LNCaP”.

We have included the results of a human Stem Cell Transcription Factor qPCR array analysis, and described the results on page 10, lines 12-15 that “We next performed a qPCR array analysis to examine the expression level of human stem cell transcription factors in CSCs and parent cells of PC3. Among the 88 transcription factors analyzed, GATA3, SOX2, and MYC were the three transcription factors that displayed elevated expression in CSCs as compared to parent cells”.

4. How reliable are basal and luminal markers analysis used in the study to determine the differentiated phenotype? The previous studies showed that CRPC CSC ALDH+/CD44+/α2β1+ cells have a high expression of basal cell genes (e.g. CK5, CK14) whereas these markers were not detected in LNCaP cells (Tang DG et al, Clin Cancer Res 2016; Oncotarget 2015).

Response: We have changed the term “differentiation markers” to “basal, luminal, and neuroendocrine markers” in the entire manuscript to more truly reflect the properties of these markers. The finding that the expression of CK5, CK14, and CK18 was universally decreased in CSCs cloned from PC3, DU145, and LNCaP cells was discussed in conjunction with the previous findings suggested by the reviewer on page 8, lines 3-7. The two references recommended by the reviewer have been cited as Ref 38 and 43, respectively.

5. The authors are encouraged to characterize a potential correlation of ANG and PLXB2 expression with patients’ clinical outcome in the publicly available prostate cancer gene expression data sets.

Response: We have cited 7 references that report upregulation of ANG in prostate cancer as well as an inverse correlation of patient survival with PLXNB2 expression on page 10, bottom two lines and page 11, top two lines, where it reads: “ANG has been shown to be progressively up-regulated in prostate cancer^{8,16,17,19,45} and plays a role in the transition from androgen-dependent to castration-resistant, hormone-refractory phenotype^{21,45}. PLXNB2 has also been shown to be up-regulated in prostate cancer and be inversely correlated with patient survival¹⁴”.

6. Did the authors use Isotype IgG as control treatment for the experiment on Figure 4 A?

Response: Yes, the control was a non-immune isotype IgG. The labels in Figure 4a as well as in Supplementary Figures 7a and 7b have been changed.

7. How did the authors control for the specificity of the IHC staining?

Response: We described on page 36, lines 2-3 that “Endogenous enzymes were blocked with 0.3% hydrogen peroxide in TBS for 15 min at RT”, and on lines 7-8 that “The specificity of primary antibodies was confirmed by immunoblotting. The specificity of secondary antibody was confirmed by replacing primary antibody with an isotype IgG control”.

8. The authors need to report core information about the antibodies that were used in their studies including the antibody name, the company, the catalogue/clone numbers.

Response: We have added catalog number of every commercial antibody in the Method section. This information is also provided in Reporting Summary.

9. The manuscripts would substantially benefit from a careful proofreading.

Response: The manuscript has been carefully proofread.

Reviewer #2

The manuscript by Hu and colleagues addressed the question how prostate cancer stem cells (CSC) survive chemotherapy. They discovered a novel mechanism how angiotensin is regulating prostate CSC survival via plexin-B2 and 5S RNA. Within a previous publication from same group of Hu they discovered the connection how the ANG-Plexin-B2 pathway is regulation cell growth and proliferation via the AKT/ERK pathway, rRNA transcription and ribosomal biogenesis and via tRNAs under physiological and pathophysiological conditions (Yu et al. 2017, Cell). Other groups demonstrated that in prostate cancer the oncogene c-myc is regulating cell cycle arrest, differentiation and apoptosis via the 5S RNP complex and mdm2/p53. The role of the ANG-PLXNB2-5S RNA for prostate CSC survival is not known so far and is in detail analyzed within the submitted manuscript. Despite the manuscript is well written, easy to follow and the figures are illustrative, there are some points, which should be addressed before publication in Communications Biology.

Response: We thank the reviewer for the positive comments and enthusiasm.

1. Within the introduction you are stating on page 4 that no reliable prostate CSC marker exist. This is not true, despite I agree that from most of the used marker functional testing is missing. But established markers, such as CD44+, CD133+, integrins, ALDH1A1, ...and should be at least mentioned.

Response: We now stated on page 4, lines 6-9, that “Numerous cell surface markers including aldehyde dehydrogenase (ALDH), CD133, CD44, CD24, and CD49f (ITGA6B), have been reported to be associated with prostate CSCs³⁰⁻³³. However, until now, reliable cell surface markers that can be used to sort authentic prostate CSCs are still lacking³⁴” to emphasize that no reliable markers for sorting CSCs.

2. Regarding the single cell experiments, please indicate how many clones were analyzed, what was the clonal distribution and what the clonal heterogeneity. Did you observe differences within the 3 used cell lines PC3, DU145 and LNCaP? (Fig. 1b)

Response: In the revision, we described on page 5, lines 5-8 that “A total of 18 clones (6 each from PC3, DU145, and LNCaP) were examined for their ability to initiate xenograft tumors in NSG mice. Among them, two clones from PC3 and one clone each from DU145 and LNCaP were able to form tumors in NSG mice with 100 cells and were thus further studied”. We also described on page 5, lines 10-16 that “Two of the six mice inoculated with a single CSC1 of PC3 developed tumors, and six of the eight mice developed tumors when inoculated with a single CSC of DU145 and LNCaP lines (Fig. 1c). No tumors were detected in mice inoculated with 1 or 10 CSC2 line of PC3 (Fig. 1c), suggesting considerable clonal heterogeneity of CSCs cloned from established cell lines. CSC1 of PC3

was used throughout the study and CSC2 was no longer studied. Under the same conditions, at least 50,000 parent PC3 cells were required to initiate xenograft tumors in NSG mice”

3. In Fig. 1d the self-renewal was analyzed from the isolated single cell clones, despite the sphere-forming capacity is twice increased compared to the parental cell line. The clones lose the self-renewal potential during *in vitro* culture. A critical discussion would be necessary. I would also suggest including limiting dilution and serial passaging analysis here.

Response: The sphere forming capacity of CSCs is actually increased by 44.6-, 53.6-, and 48.6-fold compared to parent lines, which was described on page 5, bottom two lines and on page 6, top two lines where we stated that “The number of spheres formed from 10,000 CSCs of PC3, DU145, and LNCaP was 22.3 ± 1.4 , 17.7 ± 2.1 , and 24.3 ± 3.1 , respectively, representing an increase of 44.6-, 53.6-, and 48.6-fold over that from the same numbers of the respective parent cells, which was 0.5 ± 0.1 , 0.33 ± 0.29 , and 0.50 ± 0.25 , respectively (Fig. 1d, left panel)”

We have performed limited dilution and serial passaging analysis and included the data in Fig. 1d, middle and right panels, and described the results on page 6, lines 3-6 where we stated that “Similar results were obtained in limited dilution analysis with PC3 CSCs (Fig. 1d, middle panel). No appreciable decrease in sphere-forming ability of PC3 CSCs was noted for at least 5 passages in serial re-plating experiments in sphere medium in non-adherent dishes (Fig. 1d, right panel)”.

4. In Fig. 1e the author’s shows that, the selected clones are mainly in G0 phase. Beside flow cytometry analysis, cell growth assays are necessary here. Are these cells still metabolic active? Are they senescent? Do you see differences in your 3D sphere-forming culture compared to the adherent culture? What is the potential of these quiescent cells to reactive proliferation and which rate?

Response: We have compared growth and proliferation characteristics of CSCs and parent cells in adherent culture and *in vivo*. The data were included as Fig. 1g and 1h and described on Page 6, last paragraph and on page 6, first paragraph that “Consistent with the quiescent status and a low protein synthesis rate, CSCs have reduced proliferation rate as compared to their respective parent cells. They proliferated slower *in vitro* than the parent cells until day 40 in culture (Fig. 1g) with the biggest difference observed in the early phase of culture. For example, culture of 500 PC3 parent cells for 12 days resulted in 33,150 cells, representing a 65.3-fold increase in cell number, whereas culture of 500 CSCs resulted in 2,290 cells in the same period, representing only a 3.6-fold increase in cell number (Fig. 1g). The difference in proliferation rate between CSCs and parent cells of PC3 gradually decreased in a prolonged culture and reversed by day 40, when PC3 parent cells reached a plateau but CSCs remained proliferating, suggesting an unlimited proliferation potential of CSCs, a phenomenon that has been previously proposed³⁷. Tumors initiated from CSCs also grew slower *in vivo* than did those initiated from an equal number of parent cells (Fig. 1h, inset) before they picked up speed around week 2. In the later phase of the *in vivo* growth, CSC-initiated tumors grew much faster than those derived from parent cells, resulting in 2.7-fold larger tumors (Fig. 1h). Similar growth characteristics were also observed in CSCs of DU145 and LNCaP cells (Supplementary Fig. 1a,

b). These data demonstrate that CSCs are metabolically active and are not senescent, and are able to proliferate and differentiate *in vitro* and *in vivo*".

5. In Fig. 1g the authors demonstrate bone tropism of the selected clones. Did you analyze any other organ for metastasis, e.g. lymph nodes?

Response: In the time period of this experiment (2 weeks), we did not observe lymph node metastasis. We describe this finding on page 7, lines 1-3 from bottom that "No GFP-labeled parent cells or CSCs were detected in other organs including lungs and lymph nodes in these animals under this condition (2 weeks post-transplant with 50,000 cells per mouse)".

6. For the whole manuscript the authors use CK5, CK14, CK18 and SYP/CHGA as surrogate marker for prostate CSCs. The used marker are exclusively labelling cells from the three described cell lineages within the prostate: basal, luminal and endocrine, but cannot be used to determine CSC count. Please add described markers, such as CD133, Aldefluor (ALDH1A1), Integrins, CXCR4, Nanog, Sox2, b-catenin, ... and EMT markers (in Fig. 3a).

Response: CK5, CK14, CK18, and SYP were used as surrogate markers for the differentiation status of CSCs. However, we have changed the term "differentiation markers" to "basal, luminal, and neuroendocrine markers" to indicate those are cell lineage markers. We have characterized the expression of a number of reported CSC markers including ALDH, CD 133, CD49f (ITGA6B), CD24, CD44, CD326, and Trop2. Those data were presented in Supplementary Figure 4a-4g, and were described on page 10, lines 3-11 that "Cell surface markers including ALDH, CD24, CD44, CD49f, CD133, CD326, and Trop2, have been used to sort potential prostate CSCs³⁰⁻³³. We analyzed the expression patterns of these cell surface molecules in the CSCs cloned from PC3, DU145, and LNCaP cell lines by flow cytometry and found that all three CSC lines have increased expression of CD49f (Supplementary Fig. 4a) and decreased expression of CD133 (Supplementary Fig. 4b). No consensus expression patterns of ALDH, CD24, CD44, CD326, and Trop2 were noticed among the three CSC lines (Supplementary Fig. 4c-4g). These results suggest that markers for tumorigenicity, aggressiveness, and migration are not universal CSC markers in prostate cancer".

The expression levels of other stemness-related genes including GATA3, ITGA2, MUC1, BMP7, DNMT1, KLF4, SOX2, CCLD4, MYC, p27, and p21 were presented as heatmaps in Figure 3a, 3d, 4b, 8a, and 8d.

7. In fig 2e and 2f the authors stating in the text that tumor growth pattern and histology is different in clones vs. parental cell lines. I do not see this in the figure with the used markers. Is the tumor stage different (e.g. using HE staining)?

Response: Actually, we stated that tumors derived from CSCs and parent cells have a similar level of CK18 and STP. To make this clearer, we have included higher magnification images in Figures 2e and 2f, and described the results on page 9, lines 3-5 that "As a result, tumors derived from CSCs and parent cells of PC3 displayed a similar level of CK18 (Fig. 2e) and SYP (Fig. 2f), indicating that both parent cells and CSCs were able to form differentiated tumors in mice".

8. Because Plexin-B2 is a cell membrane receptor, I was wondering if it would be possible to isolate the positive cells via FACS and compare to the negative cells concerning cell proliferation, sphere-formation, chemotherapy response and tumorigenicity.

Response: We have characterized sphere forming capacity of $PLXNB2^{high}CD49f^{high}ALDH^{high}$ cells and included the result as Supplementary Fig. 6f and described it on page 12, first paragraph that “As $PLXNB2$ (Supplementary Figure 5), $CD49f$ (Supplementary Figure 4a), and $ALDH$ (Supplementary Figure 4c) are over-expressed in CSCs than in parent cells of PC3, we sorted $PLXNB2^{high}CD49f^{high}ALDH^{high}$ cells from PC3 cells and examined their ability to form prostatospheres. However, $PLXNB2^{high}CD49f^{high}ALDH^{high}$ cells displayed no enhancement in sphere-forming capabilities as compared to the parent cells (Supplementary Fig. 6f), indicating that $PLXNB2$ is upregulated in CSCs but is insufficient to be used for sorting potential CSCs”.

9. To demonstrate the role of the ANG-Plexin pathway for cell migration, cell migration and invasion assays would be necessary upon ANG capturing.

Response: We have performed a spheroid invasion and cell proliferation experiments to examine the effect of ANG and $PLXNB2$ knockdown on cell proliferation and invasion. These results were presented in Supplementary Fig. 6a-6e, and described on page 11, lines 8-15 that “Consistent with more cells in the S-G2-M phase, ANG and $PLXNB2$ knockdown CSCs proliferated significantly faster than did control CSCs both *in vitro* (Supplementary Fig. 6a, b) and *in vivo* (Supplementary Fig. 6c). Tumors derived from ANG and $PLXNB2$ knockdown CSCs were larger than those derived from the same number of control shRNA-transfected CSCs, accompanied with a significant increase in Ki-67 positive cells in the tumor sections (Supplementary Fig. 6d). No difference was noted in spheroid invasion⁴⁶ of ANG and $PLXNB2$ knockdown CSCs into a gel comprising of collagen and laminin (Supplementary Fig. 6e), suggesting that ANG and $PLXNB2$ may not be involved in prostate CSC migration and invasion”.

10. What are the effects of the therapeutic antibodies 26-2F and mAb17 on normal tissue? Did you observe any late effect toxicities?

Response: We did not observe adverse effect of mAb17 as has been reported previously. On page 14, lines 5-7, we stated that “Consistent with previous reports that inhibition of $PLXNB2$ by mAb17 had no adverse effect on normal tissues¹⁴, no acute or delayed toxicity was noticed in mAb17-treated NSG mice”.

11. Fig. 6 is demonstrating delayed tumor growth upon DTX or mAb17 treatment, beside the CK IHC staining it would be necessary to analyze clonal composition and CSC marker expression in the recurrent tumors.

Response: We have examined the expression levels of ANG , $PLXNB2$, $CD49f$, $CD133$, $CD24$, and $CD44$ in tumor tissues derived from mice treated with DTX or with DTX + mAb17. These results were included as Fig. 6c and described on 14, lines 11-17 that “Recurrent tumors from DTX + mAb17 group had slightly enhanced ANG expression as compared to those from DXT group, probably as a

feedback response from PLXNB2 inhibition by mAb17 (Figure 6c). However, no difference was observed in recurred tumors between DTX and DTX + mAb17 groups in the expression levels of PLXNB2 and CSC markers including CD49f, CD133, CD24, and CD44 (Figure 6c), suggesting that delayed tumor recurrence in the combinatorial treatment group was not due to changes of clonal composition induced by PLXNB2 inhibition”.

Minor comment:

1. Include calculation for synergy in Fig. 6a.

Response: The synergistic effect of DTX and mAb17 has been calculated and included as Supplementary Fig. 8a and 8b. We describe on page 13, last line and on page 14, lines 1-6 that “mAb17 and DTX at 10 mg/kg each marginally inhibited CSC tumor growth (Fig. 6a), and displayed an additive effect when used together (Supplementary Fig. 8a). At a high dose (30 mg/kg), DTX had a greater inhibition and was further enhanced by mAb17, resulting in over 90% inhibition in tumor growth in the combinatorial treatment group (Fig. 6a). The degree of inhibition in the combinatorial treatment group was significantly higher than the theoretically additive value of the two individual treatment groups (Supplementary Fig. 8b), indicating a synergistic effect between mAb17 and DTX”.

2. The IHC images are too small. Please include also images with higher magnification. (Fig. 2e,f, Fig.6c,d)

Response: Higher magnification images have been added in Fig. 2e and 2f. We have also replaced the lower magnification images in Fig. 6d and 6e with high magnification ones, and moved the lower magnification images to Supplementary Figures (8c and 8d).

Reviewer #3 (Remarks to the Author):

It is very interesting that angiogenin-plexin-B2 (ANG-PLXNB2) pathway plays a critical role in the regulation of cancer stemness of prostate cancer. Furthermore, the possibility that the inhibition of the ANG-PLXNB2 pathway can conquer the chemoresistance in the prostate cancer could be benefit for future cancer treatment. Thus, this paper will be worthy to be published, but there are a few things to be modified for more confirmation.

Response: We thank the reviewer for the positive comments and enthusiasm.

1. Although the findings are novel, but it's very curious how the authors happened to find the involvement of the ANG-PLXB2 pathway in the regulation of cancer stemness. There is no description about it.

Response: We have previously reported that ANG-PLXNB2 pathway promotes stemness of hematopoietic stem cells (HSPCs) while simultaneously enhances proliferation of myeloid progenitor cells (MyePros)^{6,14}. Since CSCs and HSPCs share some similarities in regulation and since ANG-PLXNB2 has been shown to enhance proliferation of prostate cancer cells, we assumed that ANG-

PLXNB2 may also promote stemness of prostate CSCs as it does to HSPCs. This rationale is described in the Introduction section of the revised manuscript on page 3 and page 4, where we stated that “The dichotomous functions of ANG and PLXNB2 in HSPCs and MyePros prompted us to examine their function in prostate CSCs and in differentiated cancer cells, based on the rationale that CSCs and HSPCs are regulated by similar mechanisms¹⁵. We hypothesized that while ANG stimulates proliferation of differentiated prostate cancer cells, a function well-documented in the literature^{8,16-22}, it restricts proliferation of prostate CSCs and preserves their self-renewal capacities by promoting quiescence as it does to HSPCs”.

2. As you know that antibodies mostly obtained commercially have some additives, then those additives often interfere with cell growth. Thus, it is important to use control antibodies such as normal IgG. It is recognized that some experiments were done using such control antibodies, but not all. At least Fig. 4a, Fig. 5g, Fig. 6 should have control antibodies.

Response: We used isotype IgG in Fig. 4a and have changed the labels in Fig. 4a as well as in Supplementary Fig. 7a and 7b in the revised manuscript (Supplementary Figure 4a and 4b in the original manuscript).

No commercial antibodies were used in tumor therapy experiments in Fig. 5g and in Fig. 6. mAb17 was used in these experiments, which was made in house from hybridoma cells. No additive or preservative was ever added in the antibody prep. We have stated on page 20, lines 1-3 that “...which were prepared in house from hybridoma cell culture and purified to homogeneity (>99% purity shown by SDS-PAGE) containing no preservatives or additives such as sodium azide or glycerol”. We believe that vehicle (PBS) is acceptable as a proper control in tumor therapy experiment. In all in vitro experiments, isotype IgG was used as antibody control (Fig. 4, 5d, 5e, 5f, Supplementary Fig. 7), which established the specificity of mAb17 and 26-2F. We hope that the reviewer can agree with us that it is unnecessary to repeat the tumor therapy experiments shown in Fig. 6a and 6b, given that the specificity of mAb17 (made-in-house and no additives) has already been established by in vitro experiments and by previous publication (Ref. 14), and because of the large number of NSG mice (90 mice at \$172 per mouse) required for these experiments.

3. It is surprising that PC-3 and DU-145, androgen-independent cancer cells, potentially have AR expression abilities. It needs some reference papers.

Response: We have cited reference 40 and described on page 9, lines 6-8 that “AR mRNA and protein are detectable in both androgen-sensitive LNCaP cells and -insensitive DU145 and PC3 cells⁴⁰”.

REVIEWERS' COMMENTS:

Reviewer #1 (Remarks to the Author):

The authors addressed the main concerns from the reviewers; the revised version of the manuscript can be accepted for publication in Communications Biology.

Reviewer #2 (Remarks to the Author):

The authors substantially improved the manuscript, included almost all experimental data suggested by the three reviewer and answered carefully every comment from the reviewers. I do not agree with the authors response to comment 4 of reviewer 2: If the proliferation rate of CSCs is lower compared to the parental lines. Then the conclusion that parental cells stop growing around day 40 and CSCs still survive due to unlimited proliferation is of debate, because doubling times counting. If we assume that parental cells have e.g. a doubling time of 24h and would cycle 40x until day 40. How often divide the CSC cultures until day 40 and are these cells able to divide more often then 40x? It would be of advantage to at least comment on this point more carefully within the manuscript. Nevertheless, I highly recommend the recent version for publication.

Reviewer #3 (Remarks to the Author):

The authors responded to my points and revised thoroughly.

We thank the reviewers for their positive comments and enthusiasms of the revised manuscript. The following is the response to the additional comments of reviewers.

Reviewer #1

The authors addressed the main concerns from the reviewers; the revised version of the manuscript can be accepted for publication in Communications Biology.

Response: We thank the reviewer for the positive comments.

Reviewer #2

The authors substantially improved the manuscript, included almost all experimental data suggested by the three reviewer and answered carefully every comment from the reviewers. I do not agree with the authors response to comment 4 of reviewer 2: If the proliferation rate of CSCs is lower compared to the parental lines. Then the conclusion that parental cells stop growing around day 40 and CSCs still survive due to unlimited proliferation is of debate, because doubling times counting. If we assume that parental cells have e.g. a doubling time of 24h and would cycle 40x until day 40. How often divide the CSC cultures until day 40 and are these cells able to divide more often then 40x? It would be of advantage to at least comment on this point more carefully within the manuscript. Nevertheless, I highly recommend the recent version for publication.

Response: We thank the reviewer for the comments and respond as follows. We have deleted the phrase "...suggesting an unlimited proliferation potential of CSCs" (page 7, line 5 in the "track changes" version). The final version reads: "The difference in proliferation rate between CSCs and parent cells of PC3 gradually decreased in a prolonged culture and reversed by day 40, when the parent cells reached a plateau but CSCs remained proliferating, a phenomenon that has been previously observed".

Reviewer #3

The authors responded to my points and revised thoroughly.

Response: We thank the reviewer for the positive comments.